# A Deep Learning-Enabled Digital Twin Framework for Fast Online Adaptive Proton Therapy: A Validation Study in A Prostate SBRT Clinical Application

**Chih-Wei Chang**[*1] (iD)                        CHIH-WEI.CHANG@EMORY.EDU

**Mojtaba Safari**[1] (iD)                          MOJTABA.SAFARI@EMORY.EDU

**Sri Sai Akkineni**[2]                             SAKKINENI@AUGUSTA.EDU

**Mingzhe Hu**[1]                                   MINGZHE.HU@EMORY.EDU

**Keyur D. Shah**[1]                                KEYUR.DEVENDRA.SHAH@EMORY.EDU

**Pretesh Patel**[1]                                PRETESH.PATEL@EMORY.EDU

**Ashesh B. Jani**[1]                               ABJANI@EMORY.EDU

**Greeshma Agasthya**[1,3]                          GREESHMA.AGASTHYA@EMORY.EDU

**Jun Zhou**[1]                                     JUN.ZHOU@EMORY.EDU

**Xiaofeng Yang**[*1] (iD)                          XIAOFENG.YANG@EMORY.EDU

[1] *Department of Radiation Oncology and Winship Cancer Institute, Emory University, Atlanta, GA 30308,*

[2] *Department of Medicine, Medical College of Georgia, Augusta, GA, 30912,*

[3] *George W. Woodruff School of Mechanical Engineering, Georgia Institute of Technology, Atlanta, GA 30332*

**Editors:** Accepted for publication at MIDL 2026

## Abstract

Online adaptive radiotherapy offers substantial potential for improving treatment precision by accounting for daily anatomical variations, yet conventional replanning workflows remain time intensive and limit feasibility for hypofractionated treatments such as prostate stereotactic body radiation therapy (SBRT). This validation study demonstrates a deep learning enabled digital twin (DT) framework that leverages a VoxelMorph-based multi atlas deformable image registration pipeline to enable fast online adaptive proton therapy planning with dominant intraprostatic lesion (DIL) boost while achieving clinical equivalent plan quality with significantly reduced reoptimization time. The DT framework integrates deformable registration, daily cone beam CT (CBCT)-driven anatomical updates, and knowledge-based composite scoring functions, using an institutional database of 43 prostate SBRT patients with 215 CBCT scans totaling approximately 26,312 images to forecast interfractional variations and pre generate probabilistic treatment plans for new patients. Upon daily CBCT acquisition, the system enables rapid reoptimization using pre-computed plan conditions, and plan quality is evaluated using a ProKnow based scoring system that assesses target coverage and organ at risk sparing. Across all cases, the DT framework achieved an average reoptimization time of $5.5 \pm 2.7$ minutes compared with $19.8 \pm 11.9$ minutes for clinical workflows, representing a 72 percent reduction, while producing optimal plans with a composite score of $157.2 \pm 5.6$ compared with $153.8 \pm 6.0$ for clinical plans. DT generated plans maintained high dosimetric quality, including DIL V100 of 99.5 percent $\pm 0.6$ percent, CTV V100 of 99.8 percent $\pm 0.2$ percent, and comparable sparing of organs at risk, such as bladder V20.8Gy of $11.4 \pm 4.2$ cm$^3$, rectum V23Gy of

---

[*] Corresponding authors: chih-wei.chang@emory.edu, xiaofeng.yang@emory.edu

$0.7 \pm 0.4$ cm$^3$, and urethra D10 of 90.9 percent $\pm2.3$ percent. These results demonstrate that deep learning enabled digital twins can substantially accelerate online adaptive proton therapy while preserving or enhancing plan quality, providing a clinically feasible pathway toward real time personalized radiotherapy for prostate SBRT with DIL boost.

**Keywords:** Digital twins, VoxelMorph, deformable image registration, adaptive proton therapy, cone-beam CT, prostate SBRT, deep learning, medical image registration

## A. Introduction

The digital twin (DT) paradigm, initially conceptualized for aerospace engineering applications (Glaessgen and Stargel, 2012), has emerged as a transformative computational framework in precision medicine (Chaudhuri et al., 2023). Digital twins represent high-fidelity virtual replicas of physical systems that enable real-time monitoring, predictive analytics, and scenario. In the context of oncology, digital twins integrate multiphysics modeling, probabilistic simulations, and artificial intelligence to create patient-specific computational surrogates that guide clinical decision-making (Chang et al., 2025). The healthcare application of digital twins addresses the fundamental challenge of medical complexity, where symptoms, disease pathways, and treatment interactions create multidimensional decision spaces that exceed human cognitive capacity for optimal navigation (Björnsson et al., 2019; Hormuth et al., 2021). Recent advances in deep learning have accelerated the practical implementation of medical digital twins by providing computational tools capable of learning complex mappings from high-dimensional medical imaging data simulation (Katsoulakis et al., 2024). Deep learning models with extensive and growable structures have demonstrated remarkable success in tasks ranging from image segmentation to disease classification, while generative models enable synthesis of realistic medical images for simulation purposes. The convergence of these technologies with the digital twin framework creates opportunities for personalized treatment optimization that were computationally intractable. A comprehensive overview of medical digital twin methodologies and deep learning approaches for online adaptive radiotherapy is presented in Appendix A.

Prostate cancer accounts for roughly 30% of new cancer diagnoses in U.S. men (Schaeffer et al., 2024), and dominant intraprostatic lesions (DIL)—the most aggressive tumor foci—are present in approximately 83% of cases (Chen et al., 2000). Because DIL strongly correlate with local recurrence, precise targeting is essential but technically challenging due to their small volume and the prostate's susceptibility to interfractional motion from bladder and rectal filling. These issues highlight the need for approaches that accommodate daily anatomical variability. Stereotactic body radiation therapy (SBRT) with a simultaneous integrated boost to DIL offers a promising strategy, delivering highly conformal, hypofractionated treatments in few sessions (Mancosu et al., 2016), where adaptation becomes especially important. Proton therapy provides additional dosimetric advantages via the Bragg peak but is highly sensitive to anatomical changes and range uncertainties (Chang et al., 2022a,b, 2023a), further underscoring the importance of adaptive planning (Bobić et al., 2021; Paganetti et al., 2021).

Deformable image registration (DIR) represents a fundamental capability for adaptive radiotherapy (Oh and Kim, 2017), enabling the mapping of anatomical structures between imaging timepoints. Traditional DIR approaches based on iterative optimization of intensity-based similarity metrics require substantial computational time, limiting their

utility for online adaptive workflows. The emergence of deep learning-based registration methods has transformed this landscape by learning parameterized registration functions that can be applied rapidly to new image pairs without per-case optimization. Voxel-Morph (Balakrishnan et al., 2019) exemplifies the deep learning approach to medical image registration (Zhu and Lu, 2022; Feenstra et al., 2024; Chen et al., 2020), employing a U-Net encoder-decoder architecture to predict dense deformation vector fields (DVF) from input image pairs. The unsupervised learning paradigm of VoxelMorph is particularly attractive for medical applications, as it relies solely on image intensities without requiring ground truth deformation fields that are difficult to obtain in clinical settings. The framework's loss function combines an image similarity term (typically normalized cross-correlation) with a regularization term that encourages smooth, anatomically plausible deformations. Extensive validation across anatomical sites has demonstrated accuracy comparable to state-of-the-art iterative methods while achieving orders-of-magnitude speedup. The application of VoxelMorph to pelvic computed tomography (CT) and cone-beam CT (CBCT) registration for prostate cancer radiotherapy has shown promising results (van Eijnatten et al., 2021; Hemon et al., 2023; Liu and Liu, 2022; Deng et al., 2023), with recent studies demonstrating effective contour propagation and dose accumulation capabilities. The framework's flexibility allows incorporation of auxiliary information such as anatomical segmentations during training to further improve registration accuracy for specific clinical applications. These characteristics make VoxelMorph an ideal component for digital twin frameworks requiring rapid, accurate anatomical mapping.

This validation study presents a deep learning-enabled digital twin framework for online adaptive proton therapy, demonstrating its clinical feasibility through comprehensive evaluation on institutional prostate SBRT data. The framework integrates VoxelMorph-based multi-atlas deformable image registration, knowledge-based plan quality evaluation, and rapid online reoptimization capabilities. Our specific contributions include: 1) A multi-atlas registration strategy that leverages population-level motion patterns to predict patient-specific anatomical variations, generating comprehensive libraries of predicted CT images for pre-treatment planning; 2) Integration of VoxelMorph-based deformable image registration into a digital twin workflow enabling rapid online plan selection and reoptimization based on daily CBCT imaging; 3) Validation demonstrating 72% reduction in reoptimization time ($5.5 \pm 2.7$ minutes vs. $19.8 \pm 11.9$ minutes) while maintaining clinical-equivalent plan quality as assessed by composite scoring functions; 4) Comprehensive dosimetric analysis confirming adequate target coverage (DIL V100 $\geq$ 99.5%) and organ-at-risk sparing comparable to clinical standards.

Recent work has demonstrated that machine learning driven acceleration of radiotherapy planning can substantially reduce clinical workload while maintaining plan quality. In a prospective clinical deployment study, McIntosh et al. (2021) showed that machine learning based prostate radiotherapy planning reduced the median end to end planning time by more than 60 percent compared to conventional workflows, while achieving high clinical acceptability and physician selection rates. Importantly, this study emphasized that planning time reduction is a clinically meaningful objective only when algorithms are fully integrated into real world clinical workflows and evaluated under prospective conditions. These findings provide strong clinical motivation for further efforts to accelerate adaptive radiotherapy

workflows, particularly for time sensitive scenarios such as online adaptive proton SBRT, where treatment decisions must be made within a strict on couch time window.

## B. Methods

**Digital Twin Framework Architecture**  The proposed digital twin framework (Figure B1) for online adaptive proton therapy comprises three integrated modules: (1) a multi-atlas DIR module for generating predicted CT images representing potential anatomical variations, (2) a clinical proton treatment workflow module managing initial planning and online dose evaluation, and (3) a knowledge-based plan quality evaluation module enabling systematic assessment of treatment plans. The framework architecture is designed to support both offline pre-treatment preparation and online treatment-day adaptation, with the digital twin concept embodied in the patient-in-silico model that anticipates anatomical variations before treatment delivery.

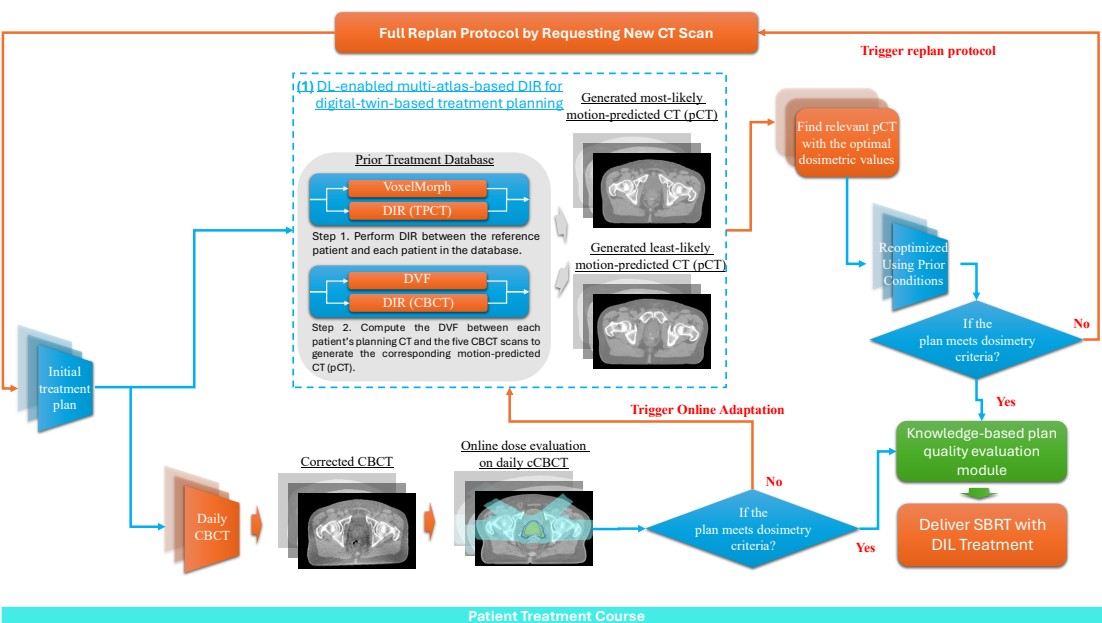

Figure B1: Digital twin framework for fast proton online treatment adaptation. The treatment planning incorporates predicted interfractional anatomical variations through predicted CT image sets generated by DL-based DIR, enabling rapid and personalized plan adaptation.

During the offline phase, the framework processes the patient's treatment planning CT (TPCT) through the multi-atlas registration module, which includes two deformation steps. In the first step, conduct DIR between the selected reference patient and all patients in the database to capture inter-patient anatomical variations and derive mappings that reflect population-level motion and deformation patterns. VoxelMorph-based deformable image registration aligns prior TPCTs from the institutional database to the current patient's anatomy, with image similarity quantified using composite evaluation metrics. The top-

performing prior patients, those with the highest similarity to the new patient, are selected to provide motion correlation information from their corresponding CBCT series. This selection strategy is based on the hypothesis that biomechanical motion patterns follow physiological constraints, enabling transfer of population-level motion information to new patients with similar anatomical characteristics. In the second step, perform deformable image registration between the planning CT and each of the five CBCT acquisitions to obtain DVF. These DVFs are then applied to the planning CT to generate the motion-predicted CTs (pCT), representing the patient's anatomy under different treatment-day conditions. The deformation vector fields derived from similar prior patients are applied to the current patient's TPCT to generate predicted CT images representing potential interfractional anatomical variations. Treatment plans are computed on both the original TPCT (clinical approach) and the generated pCT images (digital twin approach), creating a library of adaptation options for online use. This pre-computation strategy eliminates the need for time-consuming de novo plan optimization during online workflows, instead enabling rapid selection and refinement of pre-computed plans.

During the online phase, daily CBCT imaging provides the current anatomical state of the patient in treatment position. The clinical workflow module generates corrected CBCT (cCBCT) images with Hounsfield units consistent with TPCT, enabling accurate dose calculation. Online dose evaluation assesses the current treatment plan against the daily anatomy, identifying cases where dosimetric parameters deviate from planning conditions. When deviations exceed tolerance thresholds, the knowledge-based evaluation module scores available plan options, and the digital twin plans enable rapid reoptimization using pre-established planning conditions. Detailed treatment planning protocol is given in Appendix B, and the evaluation metrics of image quality are given in Appendix C.

**Patient-Specific Model Instantiation Using Population Priors**  The framework employs a patient-specific training strategy designed to leverage population-level anatomical priors while avoiding information leakage. The DIR module is based on an unsupervised VoxelMorph formulation instantiated separately for each evaluated patient. For a given "current" patient, the VoxelMorph model is trained exclusively using planning CT images from prior patients as moving images, with the current patient's planning CT as the fixed target. This strategy learns how population anatomies deform toward the current patient's anatomy without using same-patient image pairs. For each of the 43 patients in this study, 42 independent patient-specific DIR models are trained using the remaining cohort, constituting a leave-one-patient-out validation at the system level. At clinical deployment, a new patient's planning CT (unseen data) serves as the fixed reference to instantiate a patient-specific digital twin on-the-fly, enabling synchronization of prior patients' CBCTs with the current anatomy for daily CT prediction and dosimetric evaluation.

**VoxelMorph-Based Multi-Atlas Registration Strategy**  The VoxelMorph registration model employs a U-Net encoder-decoder architecture optimized for 3D medical image registration. The encoder pathway consists of convolutional layers with progressively increasing feature channels and strided convolutions for spatial downsampling. The VoxelMorph network architecture and loss function are given in Appendix D. The multi-atlas registration strategy leverages population-level information to predict patient-specific anatomical variations. The process comprises five sequential steps: 1) Registration of Historical

TPCTs: All prior TPCTs in the institutional database (26,312 images) are registered to the new patient's TPCT using the trained VoxelMorph model, producing deformed prior TPCT images aligned to the new patient's coordinate system; 2) Computation of Similarity Scores: Composite image similarity scores are computed between each deformed prior TPCT and the new patient's TPCT, combining SSIM, NCC, and LPIPS metrics with equal weighting; 3) Selection of Top Atlases: The top 20% of prior patients based on composite similarity scores are selected as atlas patients, representing individuals with anatomical characteristics most similar to the new patient; 4) Transfer of Motion Patterns: For each selected atlas patient, deformation vector fields from TPCT-to-CBCT registrations across all treatment fractions are transformed to the new patient's coordinate system, capturing interfractional motion patterns; 5) Generation of pCT Library: The transferred deformation fields are applied to the new patient's TPCT, generating $n \times m$ pCT images (where $m$ is the number of CBCT fractions per atlas patient), representing predicted anatomical variations for treatment planning.

**Knowledge-Based Plan Quality Evaluation** Knowledge-based planning systems leverage accumulated clinical experience to guide treatment plan optimization and evaluation (Shah et al., 2025). The ProKnow scoring system (Nelms et al., 2012) represents one approach to knowledge-based evaluation, employing scoring functions that map dosimetric parameters to quality scores based on clinical guidelines and institutional experience. Scoring functions can be customized to specific clinical protocols (Sweat et al., 2016), such as the two-fraction prostate SBRT approach investigated in this study. In this work, the plan quality assessment employs a ProKnow-based scoring system comprising nine scoring functions adapted from clinical trial parameters for two-fraction prostate SBRT with DIL boost. Each scoring function maps a dosimetric parameter to a quality score, with higher scores indicating better plan quality. The scoring functions are designed to evaluate target coverage (DIL and clinical target volume V100), bladder sparing (V14.6Gy, V20.8Gy), rectum sparing (V13Gy, V17.6Gy, V23Gy), and urethra protection (D0.03cc, D10). The total plan quality score is computed as the sum of individual scoring function outputs, providing a single metric for plan comparison. This composite approach enables rapid identification of optimal plans from multiple adaptation options while ensuring that selected plans satisfy all relevant clinical constraints. The scoring functions can be updated based on institutional experience, enabling continuous refinement of quality assessment criteria.

## C. Dataset

**Clinical Data Collection** This retrospective validation study employed institutional CBCT image data acquired from prostate SBRT patients treated at our facility. The study cohort comprises 43 patients who underwent five-fraction prostate SBRT, with five patients receiving DIL boost treatment. Each patient underwent daily cone-beam imaging using the Varian CBCT system integrated with the ProBeam proton therapy platform, resulting in 215 CBCT volumes (43 patients $\times$ 5 fractions) for framework development and validation. There were also 43 planning CT volumes. CBCT images were acquired at 125 kVp and 176 mA with Ram-Lak kernel reconstruction, yielding volumes with $1.0 \times 1.0 \times 2.0$ mm$^3$ voxel resolution and 104 axial images per volume. Each CBCT image set included the dimensions of $512 \times 512 \times 104$ voxels. The total dataset comprises approximately 26,312

images (253 volumes × 104 images), supplemented by treatment planning CT acquisitions for each patient. Planning CT images were acquired using a Siemens SOMATOM Definition Edge scanner, providing high-quality reference images for initial treatment planning and framework training.

**Contour Delineation and Quality Assurance** Radiation oncologists meticulously delineated contours for the clinical target volume (CTV), DIL, and organs at risk (bladder, rectum, urethra) on both CT and CBCT images. Contour quality was verified through systematic review to ensure accuracy and consistency across the dataset. The DIL was identified based on multiparametric MRI findings registered to planning CT, with boundaries verified against histopathological information where available. The prescription dose of 26 Gy to the CTV with simultaneous DIL boost to 32 Gy was adapted from the 2SMART clinical trial (NCT03588819), representing current clinical practice for two-fraction prostate SBRT. Organ-at-risk dose constraints including bladder V14.6Gy $< 25$ cm$^3$, bladder V20.8Gy $< 10$ cm$^3$, rectum V13Gy $< 7$ cm$^3$, rectum V17.6Gy $< 4$ cm$^3$, rectum V23Gy $< 1$ cm$^3$, and urethra constraints were derived from the same clinical trial.

## D. Results

**Image Registration Performance** The multi-atlas registration strategy successfully identified anatomically similar prior patients and generated pCT images representing predicted interfractional variations. Composite image similarity scores (Table C1 in Appendix C) between pCT images and actual daily CBCT ranged from 0.65 to 0.88, with the highest-similarity pCT (DT-H) achieving mean composite scores of $0.81 \pm 0.05$ compared to $0.74 \pm 0.06$ for clinical TPCT. This improvement in anatomical matching demonstrates the value of population-based motion prediction for adaptive planning. Detailed analysis of image similarity metrics revealed consistent improvements across all evaluation criteria. SSIM values for DT-H pCT images averaged $0.87 \pm 0.04$ compared to $0.82 \pm 0.05$ for TPCT, indicating better preservation of structural image content. NCC values showed similar trends with DT-H averaging $0.91 \pm 0.03$ versus $0.86 \pm 0.04$ for TPCT. LPIPS perceptual similarity scores confirmed these findings, with DT-H images showing approximately 15% improvement in perceptual similarity to daily CBCT compared to TPCT.

**Plan Quality Before Reoptimization** Initial dose evaluation on daily CBCT revealed significant dosimetric deviations for clinical plans in several cases. For treatment fraction 1, clinical plans showed at least one dosimetric parameter exceeding tolerance constraints for all five test patients, indicating the need for online adaptation. Most notably, DIL V100 coverage fell below 90% for Patients 2 and 4, representing critical failures in DIL boost delivery that could compromise treatment efficacy.

In contrast, DT-based plans with high anatomical similarity (DT-H) demonstrated better initial dosimetric alignment with daily anatomy. For Patients 3 and 5, DT-H plans satisfied all evaluation criteria (DIL V100 > 95%, CTV V100 > 95%, OARs within constraints) even before reoptimization, demonstrating the potential for direct plan selection without time-consuming adaptation in favorable cases. This finding suggests that accurate anatomical prediction can substantially reduce the computational burden of online adaptive workflows.

| | | CB1 REopt | | CB2 REopt | |
|---|---|---|---|---|---|
| | | REopt Time (min) | Plan Quality Score | REopt Time (min) | Plan Quality Score |
| P1 | DT-H-REopt-A | 9.0 | 154.3 | 3.2 | 154.1 |
| | DT-L-REopt-A | 9.1 | 150.8 | 3.0 | 133.4 |
| | DT-L-REopt-B | 19.0 | 150.4 | 8.5 | 148.6 |
| | Clinic-REopt-A | 9.7 | 150.0 | 3.9 | 153.1 |
| | Clinic-REopt-B | 28.0 | 151.5 | 22.2 | 149.4 |
| P2 | DT-H-REopt-A | 8.3 | 147.2 | 6.6 | 151.8 |
| | DT-L-REopt-A | 7.8 | 145.5 | 6.8 | 151.4 |
| | DT-L-REopt-B | 31.3 | 147.6 | 14.8 | 151.8 |
| | Clinic-REopt-A | 9.0 | 142.3 | 6.7 | 150.7 |
| | Clinic-REopt-B | 46.0 | 148.2 | 26.5 | 151.2 |
| P3 | DT-H-REopt-A | 4.7 | 164.0 | 9.7 | 158.6 |
| | DT-L-REopt-A | 4.6 | 162.9 | 10.5 | 155.4 |
| | DT-L-REopt-B | 16.4 | 163.1 | 28.1 | 158.5 |
| | Clinic-REopt-A | 4.5 | 158.6 | 9.5 | 153.0 |
| | Clinic-REopt-B | 18.1 | 157.8 | 19.9 | 153.0 |
| P4 | DT-H-REopt-A | 2.4 | 165.9 | 3.3 | 160.0 |
| | DT-L-REopt-A | 2.4 | 113.4 | 3.2 | 160.1 |
| | DT-L-REopt-B | 6.4 | 164.5 | 7.7 | 159.4 |
| | Clinic-REopt-A | 2.9 | 161.1 | 3.6 | 158.3 |
| | Clinic-REopt-B | 9.8 | 166.5 | 9.1 | 160.5 |
| P5 | DT-H-REopt-A | 3.8 | 156.2 | 4.0 | 159.4 |
| | DT-L-REopt-A | 3.6 | 149.3 | 4.0 | 154.2 |
| | DT-L-REopt-B | 6.7 | 152.4 | 7.3 | 154.4 |
| | Clinic-REopt-A | 2.9 | 141.5 | 5.2 | 144.0 |
| | Clinic-REopt-B | 6.1 | 147.7 | 12.2 | 152.6 |

Table D1: Reoptimization (REopt) time and plan quality comparison between conventional clinical workflow and the proposed digital twin approach, evaluated using daily CBCT at fraction 1 (CB1) and fraction 2 (CB2). Underlined values denote the highest plan quality score achieved for each patient.

**Online Reoptimization Efficiency** Table D1 depicts that reoptimization using daily CBCT images demonstrated substantial efficiency advantages for DT-based plans. We evaluated five reoptimization strategies per patient: DT-H-REopt (digital twin plan using the highest-similarity pCT), DT-L-REopt (using the lowest-similarity pCT), and Clinic-REopt (clinical plan derived from TPCT), each with suffix "-A" denoting completion within the 10-minute online adaptive threshold or "-B" indicating extended optimization time to achieve comparable quality. The DT-H-REopt-A strategy achieved clinical-equivalent plan quality (mean score $157.2 \pm 5.6$) in an average of $5.5 \pm 2.7$ minutes, comfortably within the threshold considered feasible for online adaptive proton therapy where patient positioning stability and comfort are critical considerations.

Comparison with clinical reoptimization (Clinic-REopt) revealed a 72% reduction in planning time. Clinical plans required an average of $19.8 \pm 11.9$ minutes to achieve comparable plan quality scores ($153.8 \pm 6.0$), with several cases exceeding 30 minutes. The time savings arise from two factors: (1) better initial dosimetric alignment due to anatomical similarity between pCT and daily CBCT, requiring fewer optimization iterations for convergence; and (2) preservation of validated planning conditions from the pre-treatment phase, avoiding the need for de novo constraint specification. Plans derived from lower-similarity pCT images (DT-L-REopt) required intermediate reoptimization times ($14.6 \pm 9.1$ minutes) to achieve comparable quality ($155.1 \pm 6.0$), confirming the relationship between anatomical matching and optimization efficiency. This finding validates the multi-atlas selection strategy, demonstrating that patient-specific atlas selection based on composite similarity scores effectively identifies optimal planning substrates for rapid adaptation.

**Dosimetric Outcomes** Table E2 and Table E3 (Appendix E) show that final reoptimized DT-based plans achieved excellent target coverage with DIL V100 of $99.5\% \pm 0.6\%$ and CTV V100 of $99.8\% \pm 0.2\%$, meeting or exceeding clinical requirements across all test cases. These coverage levels are consistent with prescription goals and demonstrate the framework's ability to maintain treatment efficacy despite the reduced optimization time.

The results also indicate that organ-at-risk sparing was maintained within clinical constraints, with bladder V20.8Gy of $11.4 \pm 4.2$ cm$^3$ (constraint: $< 10$ cm$^3$), rectum V23Gy of $0.7 \pm 0.4$ cm$^3$ (constraint: $< 1$ cm$^3$), and urethra D10 of $90.9\% \pm 2.3\%$ (constraint: $< 95\%$). While bladder V20.8Gy slightly exceeded the nominal constraint, the absolute volumes remained clinically acceptable and comparable to clinical plan outcomes. Figures D2 and Figure E2 depict critical perspective on the quantitative efficiency data through comparative dose-volume histogram (DVH) analysis. DVH analysis confirmed similar dosimetric profiles between DT-based and clinical plans following reoptimization, validating the clinical equivalence of the accelerated workflow. Under unconstrained optimization conditions, all evaluated methodologies—DT-H, DT-L, and clinical reoptimization—demonstrate convergence toward dosimetrically equivalent solutions. The overlapping DVH envelopes for both target structures and organs at risk confirm that final plan quality is independent of the initial planning substrate. These observations establish that the fundamental distinction among approaches is not the quality ceiling but rather the trajectory toward it: DT-H reaches acceptable quality faster, while DT-L and clinical methods require additional optimization cycles to compensate for greater anatomical mismatch between the planning image and daily anatomy.

**Patient-Specific Registration Performance**   Detailed analysis of registration performance reveals consistent improvements across patients, though with some inter-patient variability reflecting differences in anatomical complexity and motion magnitude. Patient 2 (Figure E3 in Appendix E) presented the most challenging case with substantial bladder filling variation (volume increase from 226 cc to 340 cc between planning and first treatment fraction). Despite this significant anatomical change, the multi-atlas strategy successfully identified prior patients with similar motion patterns, generating pCT images that captured the bladder expansion. The DT-H plan achieved adequate coverage following reoptimization, though requiring longer optimization time (7.8 minutes) due to the magnitude of required plan adjustments. Patient 4 (Figure D3) demonstrated the importance of atlas selection, with the highest-similarity atlas achieving substantially better initial plan quality than lower-ranked atlases. Analysis of atlas patient characteristics revealed that the selected atlas exhibited similar prostate position relative to pelvic landmarks and comparable rectum filling patterns, suggesting that these anatomical features drive motion correlation effectiveness.

## E. Discussion

**System-Level Innovation and Clinical Workflow Integration**   The central contribution of this work lies not in proposing a new deformable image registration algorithm, but in the system-level operationalization of learning-based registration within a digital twin framework for adaptive proton SBRT. Although VoxelMorph itself is a known model, its use here differs fundamentally from most prior applications focused on intra-patient registration or relatively rigid anatomies. This study demonstrates that learning-based deformable registration can be reliably applied to inter-patient pelvic anatomy, where anatomical variability is larger and less constrained, and where registration accuracy must satisfy stringent proton dosimetric requirements rather than image similarity alone. Within the proposed framework, the learning component enables population-level deformation priors,

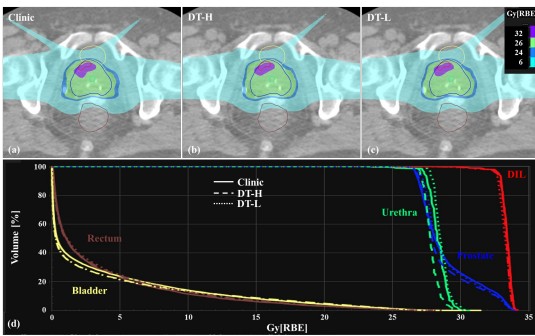

Figure D2: Dosimetric evaluation on daily CBCT for Patient 1 comparing clinical, DT-H, and DT-L reoptimized plans. DT-H and DT-L utilize pCTs with highest and lowest similarity scores to daily CBCT, respectively. Color wash dose distributions for (a) clinical, (b) DT-H, and (c) DT-L plans. (d) DVH comparison.

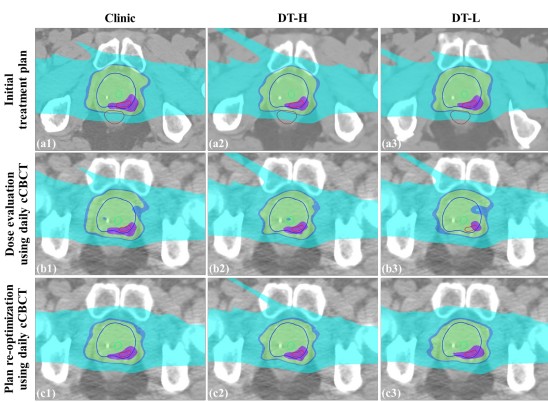

Figure D3: Dose distribution comparison for Patient 4 across treatment phases: (a1–a3) initial planned doses, (b1–b3) forward-calculated doses on daily CBCT, and (c1–c3) reoptimized doses. Columns correspond to clinical (1), DT-H (2), and DT-L (3) planning strategies.

a persistent atlas-based representation of anatomical variability, and rapid synchronization between daily CBCT and precomputed patient states. These capabilities align with formal definitions of medical digital twins that emphasize closed-loop integration of data, virtual representation, synchronization, and clinically interpretable decision interfaces, rather than isolated algorithmic novelty.

Clinical feasibility is supported by design choices prioritizing robustness, interpretability, and workflow efficiency. Image similarity metrics and atlas preselection serve only as a computationally efficient screening step to reduce adaptive planning workload, while final plan selection is governed by dosimetric evaluation using a composite ProKnow score that reflects physician-defined priorities balancing target coverage and organ-at-risk sparing. This approach mirrors real-world clinical decision making, where maximizing target dose alone is insufficient and excessive normal tissue dose is a common trigger for replanning. The framework separates pre-treatment and treatment-day operations: computationally intensive steps (deformable registration, atlas construction, population model preparation) are performed prior to treatment, while online adaptive planning is limited to plan evaluation, selection, and dose calculation within a strict on-couch time window. The reported reoptimization time of approximately 5.5 minutes reflects this clinically relevant online phase and excludes pre-treatment steps by design. This represents a 72% reduction in reoptimization time, falling within the 10-minute threshold generally considered acceptable for online adaptation (Güngör et al., 2021; McComas et al., 2023).

We acknowledge that this study does not benchmark VoxelMorph against newer registration architectures or conventional methods, reflecting the work's scope as a system-level validation rather than an algorithmic comparison. In proton radiotherapy, where planning complexity is dominated by physical interactions (range uncertainty, Bragg peak sensitivity, tissue heterogeneity), the principal translational bottleneck is not marginal gains in regis-

tration accuracy, but the ability to generate clinically acceptable adaptive plans within a strict treatment-day time constraint. The framework is designed to be registration-agnostic, allowing future incorporation of advanced models (transformer-based, segmentation-guided) without altering the core system architecture. By focusing on workflow integration, dosimetric validity, and clinical decision logic, this work addresses a critical gap in adaptive proton therapy.

**Clinical Assessment and Deployment**  The proposed digital twin framework was intentionally designed to operate under realistic clinical constraints, including contour uncertainty, decision latency, and limited computational resources during treatment delivery. Regarding contour uncertainty, the framework does not assume perfect segmentation at treatment time. The uncertainty is implicitly absorbed through population-based multi-atlas deformation, rather than relying on a single propagated contour. Plan selection is driven by robust dosimetric evaluation on CBCT-derived anatomy, rather than contour-specific optimization. Importantly, the knowledge-based composite scoring system (ProKnow) acts as a clinical safeguard, prioritizing plan robustness over marginal dosimetric gains. This reflects how digital twins are increasingly positioned as decision-support systems rather than autonomous controllers (Katsoulakis et al., 2024). Although the current study is retrospective, the system architecture addresses these prospective challenges by separating computationally intensive operations from time-critical workflows. Daily CBCT updates synchronize a pre-constructed virtual patient representation, enabling rapid decision-making at treatment time without requiring full online replanning. Contour uncertainty is managed through population-based multi-atlas deformation and robust dosimetric evaluation on CBCT-derived anatomy. Plan selection is driven by a clinically grounded knowledge-based composite metric, positioning the digital twin as a clinical decision support system rather than an autonomous controller, consistent with emerging perspectives in the digital twin literature.

From a deployment standpoint, the framework is architecturally generalizable and computationally tractable for routine clinical use. Core components (offline atlas generation, online CBCT-driven synchronization, adaptive plan library evaluation, interpretable decision logic) are reusable across disease sites, while site-specific elements are confined to data instantiation layers. The most computationally intensive steps (multi-atlas deformable image registration, deformation field generation) are performed offline prior to the first treatment fraction using GPU acceleration (NVIDIA RTX 4090 GPU in this implementation). Online adaptive planning and dose calculation were conducted within the RayStation treatment planning system using standard clinical hardware (NVIDIA Quadro RTX 8000 GPU for Monte Carlo dose calculation). Online treatment-day operations are limited to CBCT processing, adaptive plan evaluation, and plan selection, with reoptimization times of approximately 5 to 6 minutes satisfying the 10-minute on-couch clinical requirement. Storage requirements scale linearly with atlas cases, with the patient-specific predicted CT library requiring approximately 44 GB of storage per patient—a manageable overhead within existing institutional systems.

**Comparison with Alternative Approaches and Future Directions**  The digital twin framework differs from alternative adaptive radiotherapy approaches in its emphasis on predictive pre-computation rather than real-time optimization. Fully automated

planning systems based on deep reinforcement learning have demonstrated impressive capabilities for photon-based treatments, but current implementations require optimization times of 1-2 hours that are incompatible with online workflows. The pre-computation strategy employed by our framework circumvents this limitation by performing computationally intensive optimization offline, reserving only rapid refinement for the online phase. Compared to mechanistic digital twin approaches that model individual patient physiology, the data-driven multi-atlas strategy offers practical advantages for clinical implementation, as mechanistic models require patient-specific parameter calibration that may not be feasible within typical treatment preparation timelines. The framework's performance compares favorably with reported adaptive planning times of approximately 45 minutes for CT-based or MRI-guided adaptive therapy (Güngör et al., 2021; McComas et al., 2023), though direct comparison is complicated by differences in imaging modality, treatment site, and plan complexity.

Several limitations of the current study warrant consideration. The validation cohort was derived from a single institution, which may limit generalizability; multi-institutional validation with larger cohorts will be essential. The retrospective study design does not fully capture prospective clinical implementation complexities (contour propagation uncertainty, setup variability, real-time decision-making pressures). Prospective validation studies incorporating time–motion analysis and formal clinical workflow assessment will be necessary. The current framework focuses on interfractional anatomical variation using daily CBCT and does not explicitly model intrafraction motion during treatment delivery. Intrafraction dynamics, including patient motion and transient anatomical changes such as bowel gas, cannot be reliably inferred from static CBCT alone and would require integration with real-time imaging and motion-management systems (Chang et al., 2024b; Pan et al., 2025). While such capabilities are beyond the present scope, the digital twin framework is designed to accommodate these extensions. Additional limitations include the current focus on prostate SBRT; extension to other sites will require site-specific validation.

Future directions include incorporation of uncertainty quantification to identify low-confidence predictions and integration of biological response indicators for biologically informed adaptive treatment, while preserving clinical practicality through the modular architecture. For long-term deployment, several controlled mechanisms can be incorporated to improve performance as additional data become available: test-time adaptation with strict safeguards (limited patient-specific refinement under controlled conditions with pre-defined stopping criteria), institution-level continual learning (periodic offline retraining with new clinical data under formal QA workflows), and uncertainty-aware monitoring to trigger conservative fallbacks or targeted offline model updates. These approaches would enable gradual performance improvement while preserving clinical safety, reproducibility, and regulatory compliance.

## F. Conclusion

This study validates a deep learning-enabled digital twin framework for fast online adaptive proton therapy in prostate SBRT with DIL boost. The framework integrates multi-atlas deformable image registration to predict interfractional anatomical variations, enabling the pre-computation of adaptation options to accelerate online planning workflows. The key

insight is that population-level motion patterns provide a valuable prior for predicting individual patient variations. The feasibility of generating clinical-quality adaptive plans within 10 minutes suggests this framework could enable the practical implementation of online adaptive proton therapy, contributing to personalized, adaptive radiation treatment.

## Acknowledgments

This research is supported in part by the National Institutes of Health under Award Numbers R01CA215718 and R01CA272991.

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

## Appendix A. Related Work

**Deep Learning for Medical Image Registration** The application of deep learning to medical image registration has evolved rapidly since the introduction of convolutional neural network-based approaches (Balakrishnan et al., 2018; Fu et al., 2020; Krishna et al., 2021). VoxelMorph, introduced by Balakrishnan et al. (2019), established a foundational framework for unsupervised diffeomorphic registration that has been widely adopted and extended. The architecture employs a U-Net encoder-decoder structure that processes concatenated fixed and moving image pairs to predict dense displacement fields. Subsequent developments have expanded the VoxelMorph framework in several directions (Dalca et al., 2019). For pelvic imaging specifically, deep learning registration methods have demonstrated effectiveness for CT-to-CBCT alignment in prostate cancer radiotherapy (Rusanov et al., 2022). Studies (Liang et al., 2021; Elmahdy et al., 2019) have shown that VoxelMorph-based approaches achieve Dice similarity coefficients exceeding 0.85 for prostate segmentation while reducing computation time from minutes to seconds. The integration of auxiliary segmentation information during training has been shown to improve registration accuracy for specific anatomical structures of interest, making these methods particularly suitable for radiation therapy applications where target and organ-at-risk delineation is critical.

**Digital Twins in Radiation Oncology** The concept of medical digital twins has gained substantial attention in recent years, with radiation oncology representing a particularly promising application domain (Sumini et al., 2024; Zhao et al., 2025). The fundamental appeal of digital twins for radiotherapy lies in their ability to simulate treatment scenarios virtually before physical delivery, enabling optimization of treatment parameters for individual patients. Several research groups have proposed digital twin frameworks addressing different aspects of the radiotherapy workflow. Predictive digital twins for treatment optimization have been demonstrated for high-grade gliomas (Chaudhuri et al., 2023), where mechanistic tumor growth models are coupled with treatment response predictions to identify optimal radiotherapy regimens. These frameworks address uncertainties in tumor behavior through probabilistic modeling, enabling exploration of treatment alternatives that would be impractical to evaluate clinically. For prostate cancer specifically, digital twin approaches have been proposed for theranostic applications in radiopharmaceutical therapy (Abdollahi et al., 2024), addressing the challenge of optimal dosing while avoiding under- or over-treatment. The digital twin concept has also been applied to adaptive radiotherapy workflows (Chang et al., 2025), where the goal is to maintain treatment quality despite anatomical changes occurring between or during treatment fractions. Previous work (Chang et al., 2025) has demonstrated digital twin frameworks for prostate SBRT that address geometrical uncertainties and optimize dose conformity. However, the integration of deep learning-based image registration with digital twin concepts for rapid online adaptation represents a relatively unexplored area with significant potential for clinical impact.

**Adaptive Radiotherapy and Online Replanning** Adaptive radiotherapy encompasses strategies for modifying treatment plans based on anatomical or biological changes observed during the treatment course. Online adaptive workflows perform plan modifications while the patient remains in treatment position, requiring rapid computation to maintain clinical feasibility (Chang et al., 2024a). The advent of CBCT imaging on treatment machines

has enabled daily anatomical assessment, creating both the opportunity and the need for online adaptive approaches (Chang et al., 2023b). Recent advances in automated contouring (Liu et al., 2025; Carion et al., 2025) and plan optimization (Wang and Chang, 2025) have improved the feasibility of online adaptive workflows. Deep learning-based autosegmentation can reduce the time required for contour delineation from minutes to seconds, while automated planning algorithms can generate clinically acceptable plans without manual intervention. The remaining challenge lies in efficiently identifying when adaptation is necessary and selecting or generating appropriate adapted plans within the time constraints of online workflows. Our digital twin framework addresses this challenge through pre-computation of adaptation options based on predicted anatomical variations.

## Appendix B. Treatment Planning Protocol

Treatment planning employs RayStation 2023B (RaySearch Laboratories, Stockholm, Sweden) for proton therapy plan optimization and dose calculation. The treatment planning system generates corrected CBCT images by addressing scatter artifacts and Hounsfield unit inconsistencies, enabling accurate radiation dose calculation on daily imaging. GPU-accelerated Monte Carlo dose calculation ensures accurate modeling of proton interactions, particularly important for the heterogeneous tissues of the pelvic region.

Figure B1 depicts that the planning configuration employs four proton beams with gantry angles of 90°, 270°, 50° (left anterior oblique), and 310° (right anterior oblique), with beam weighting of 35%, 35%, 15%, and 15% respectively. The inclusion of anterior-oblique beams minimizes radiation exposure to the rectum while maintaining target coverage. Robust optimization addresses positional uncertainty with 5mm margins (3mm posterior) for clinical plans and 1.5mm margins for DT-based plans, reflecting the reduced uncertainty when planning on anatomy-matched pCT images. Range uncertainty of ±3.5% is incorporated through scenario-based optimization with 21 scenarios per plan.

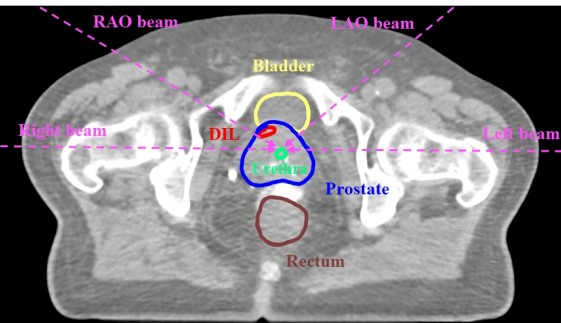

Figure B1: The beam configuration for prostate SBRT treatment plans utilizes a blend of LAO, RAO, and bilateral horizontal beams. The beam weights are allocated as follows: 15% to the LAO beam, 15% to the RAO beam, 35% to the left horizontal beam, and 35% to the right horizontal beam.

## Appendix C. Image Similarity Metrics

Comprehensive image similarity assessment employs multiple complementary metrics addressing different aspects of registration quality. Structural similarity index (SSIM) (Wang et al., 2004) evaluates perceptual image quality by comparing luminance, contrast, and structural patterns between images, providing a measure that correlates well with human visual assessment. Normalized cross-correlation (NCC) (Ayubi et al., 2024) quantifies intensity pattern similarity in a manner robust to linear intensity transformations, making it suitable for comparing images acquired with different imaging protocols.

The learned perceptual image patch similarity (LPIPS) metric (Zhang et al., 2018) leverages deep neural network feature representations to assess perceptual similarity, capturing high-level image characteristics that complement pixel-based metrics. Dice similarity coefficient evaluates the overlap between anatomical structures, providing direct assessment of registration accuracy for clinically relevant regions. Hausdorff distance at the 95th percentile (HD95) (Huttenlocher et al., 2002) quantifies boundary accuracy while remaining robust to outliers, important for evaluating organ-at-risk delineation.

The composite similarity score employed for atlas selection combines SSIM, NCC, and (1-LPIPS) with equal weighting, providing a balanced assessment of registration quality across structural, intensity, and perceptual dimensions. This multi-metric approach ensures that selected atlases exhibit high-quality alignment across multiple evaluation criteria, increasing the likelihood that transferred motion patterns will accurately predict the new patient's anatomical variations.

Table C1: Evaluation of anatomical correspondence between candidate planning images (clinical TPCT and digital twin pCT) and daily CBCT. DT-H and DT-L represent the pCT with highest and lowest composite similarity to CB1 (fraction 1) and CB2 (fraction 2).

| | Planning Image Set | SSIM | | 1 - LPIPS | | NCC | | Composite Score | |
|---|---|---|---|---|---|---|---|---|---|
| | | CB1 | CB2 | CB1 | CB2 | CB1 | CB2 | CB1 | CB2 |
| P1 | TPCT (Clinic) | 0.839 | 0.883 | 0.755 | 0.761 | 0.891 | 0.915 | 0.829 | 0.853 |
| | pCT (DT-H) | 0.865 | 0.868 | 0.771 | 0.766 | 0.895 | 0.878 | 0.843 | 0.837 |
| | pCT (DT-L) | 0.695 | 0.706 | 0.597 | 0.587 | 0.545 | 0.474 | 0.613 | 0.589 |
| P2 | TPCT (Clinic) | 0.847 | 0.839 | 0.766 | 0.787 | 0.909 | 0.919 | 0.841 | 0.848 |
| | pCT (DT-H) | 0.859 | 0.889 | 0.781 | 0.815 | 0.900 | 0.920 | 0.847 | 0.875 |
| | pCT (DT-L) | 0.677 | 0.685 | 0.606 | 0.626 | 0.660 | 0.702 | 0.648 | 0.671 |
| P3 | TPCT (Clinic) | 0.774 | 0.788 | 0.733 | 0.737 | 0.903 | 0.929 | 0.803 | 0.818 |
| | pCT (DT-H) | 0.766 | 0.779 | 0.707 | 0.724 | 0.864 | 0.870 | 0.779 | 0.791 |
| | pCT (DT-L) | 0.682 | 0.660 | 0.628 | 0.608 | 0.731 | 0.705 | 0.681 | 0.658 |
| P4 | TPCT (Clinic) | 0.719 | 0.805 | 0.665 | 0.757 | 0.826 | 0.936 | 0.737 | 0.833 |
| | pCT (DT-H) | 0.865 | 0.844 | 0.771 | 0.782 | 0.959 | 0.945 | 0.865 | 0.857 |
| | pCT (DT-L) | 0.701 | 0.666 | 0.606 | 0.581 | 0.698 | 0.744 | 0.668 | 0.664 |
| P5 | TPCT (Clinic) | 0.825 | 0.825 | 0.705 | 0.706 | 0.857 | 0.876 | 0.796 | 0.802 |
| | pCT (DT-H) | 0.824 | 0.862 | 0.705 | 0.719 | 0.796 | 0.866 | 0.775 | 0.816 |
| | pCT (DT-L) | 0.686 | 0.667 | 0.586 | 0.568 | 0.508 | 0.463 | 0.593 | 0.566 |

## Appendix D. VoxelMorph network architecture and loss function

### D.1. Network Architecture

The VoxelMorph registration model employs a U-Net encoder-decoder architecture optimized for 3D medical image registration. The encoder pathway consists of convolutional layers with progressively increasing feature channels ($32 \rightarrow 64 \rightarrow 128 \rightarrow 256$) and strided convolutions for spatial downsampling. Skip connections transfer high-resolution feature information from encoder to decoder, preserving spatial detail essential for accurate registration. The decoder pathway employs transposed convolutions for upsampling with symmetrically decreasing feature channels ($256 \rightarrow 128 \rightarrow 64 \rightarrow 32$), ultimately producing a dense 3D displacement field with three channels corresponding to displacements in x, y, and z directions.

The network processes concatenated fixed and moving image volumes, learning to predict the displacement field that aligns the moving image to the fixed image coordinate system. Leaky ReLU activations (negative slope 0.2) introduce non-linearity throughout the network, while the final layer employs no activation to allow unrestricted displacement predictions. The architecture processes 3D volumes directly using $3 \times 3 \times 3$ convolutional kernels, avoiding the information loss associated with 2D image-based approaches and ensuring spatial consistency of predicted deformations.

### D.2. Loss Function

The VoxelMorph training objective combines an image similarity term with a spatial regularization term, expressed as:

$$L = L_{\text{sim}} \left( I_{\text{fixed}}, I_{\text{moving}} \circ \varphi \right) + \lambda \cdot L_{\text{smooth}} \left( \varphi \right) \tag{1}$$

In this formulation, $I_{\text{fixed}}$ denotes the fixed (reference) image, which in our framework corresponds to the new patient's TPCT, while $I_{\text{moving}}$ represents the moving (source) image to be spatially transformed, typically a prior patient's TPCT selected from the historical atlas database. The symbol $\varphi$ denotes the DVF, a three-dimensional displacement field predicted by the network that establishes dense spatial correspondence between the moving and fixed image coordinate systems. The composition operator $\circ$ applies the predicted deformation field to warp the moving image into alignment with the fixed image, yielding the warped image $I_{\text{moving}} \circ \varphi$. The similarity term $L_{\text{sim}}$ employs normalized Cross-Correlation (NCC), which is robust to intensity variations between CT and CBCT imaging modalities. NCC measures local intensity pattern similarity within sliding windows, avoiding the assumption of identical intensity distributions that limits mean squared error-based approaches. The smoothness term $L_{\text{smooth}}$ employs a diffusion regularizer that penalizes spatial gradients of the displacement field, encouraging smooth, anatomically plausible deformations that preserve tissue topology and prevent non-physical folding or tearing of structures. The regularization weight $\lambda$ controls the trade-off between registration accuracy and deformation smoothness.

## Appendix E. Dosimetric endpoint comparison for clinical and digital twin-based plans

Table E2: Dosimetric endpoint comparison for clinical and digital twin-based plans following online reoptimization (REopt) with fraction 1 CBCT (CB1). Underlined values indicate optimal plan quality scores per patient.

| | CB1 REopt | DIL (%) V100 | CTV (%) V100 | Bladder (cc) V20.8Gy | V14.6Gy | Rectum (cc) V23Gy | V17.6Gy | V13Gy | Urethra (%) D0.03cc | D10 |
|---|---|---|---|---|---|---|---|---|---|---|
| | DT-H-REopt-A | 98.5 | 100.0 | 10.0 | 19.8 | 1.1 | 2.8 | 5.3 | 92.0 | 89.3 |
| | DT-L-REopt-A | 98.3 | 99.5 | 10.4 | 21.9 | 1.1 | 3.6 | 6.7 | 92.2 | 90.6 |
| P1 | DT-L-REopt-B | 98.1 | 99.7 | 10.0 | 21.5 | 1.1 | 3.5 | 6.5 | 94.1 | 91.6 |
| | Clinic-REopt-A | 98.1 | 99.8 | 9.7 | 22.2 | 1.2 | 3.3 | 5.9 | 93.4 | 92.2 |
| | Clinic-REopt-B | 98.1 | 99.8 | 9.3 | 21.3 | 1.2 | 3.2 | 5.8 | 91.5 | 89.2 |
| | DT-H-REopt-A | 99.8 | 99.7 | 18.6 | 33.5 | 1.0 | 3.0 | 5.4 | 94.9 | 93.8 |
| | DT-L-REopt-A | 99.3 | 99.8 | 17.8 | 32.3 | 1.1 | 3.4 | 6.5 | 94.7 | 93.5 |
| P2 | DT-L-REopt-B | 99.7 | 99.9 | 18.1 | 31.2 | 1.1 | 3.4 | 6.6 | 94.6 | 92.3 |
| | Clinic-REopt-A | 97.9 | 99.5 | 17.4 | 31.6 | 1.0 | 3.1 | 5.7 | 94.7 | 93.1 |
| | Clinic-REopt-B | 99.7 | 99.6 | 17.6 | 31.2 | 1.1 | 3.6 | 6.8 | 93.8 | 91.1 |
| | DT-H-REopt-A | 100.0 | 99.9 | 9.1 | 15.5 | 0.1 | 1.1 | 2.4 | 95.8 | 94.0 |
| | DT-L-REopt-A | 100.0 | 100.0 | 9.0 | 15.4 | 0.2 | 1.2 | 3.0 | 97.9 | 94.7 |
| P3 | DT-L-REopt-B | 100.0 | 100.0 | 9.0 | 15.5 | 0.2 | 1.2 | 3.0 | 97.8 | 94.5 |
| | Clinic-REopt-A | 100.0 | 99.9 | 11.9 | 21.5 | 0.2 | 2.1 | 4.4 | 98.2 | 95.9 |
| | Clinic-REopt-B | 100.0 | 100.0 | 11.8 | 21.5 | 0.2 | 2.1 | 4.3 | 98.6 | 96.9 |
| | DT-H-REopt-A | 99.8 | 99.2 | 9.2 | 19.4 | 0.0 | 0.4 | 1.4 | 91.1 | 88.3 |
| | DT-L-REopt-A | 84.7 | 94.9 | 11.0 | 21.8 | 0.1 | 1.0 | 2.9 | 96.9 | 92.9 |
| P4 | DT-L-REopt-B | 99.8 | 99.3 | 6.3 | 15.0 | 0.2 | 1.9 | 4.6 | 90.4 | 88.0 |
| | Clinic-REopt-A | 99.2 | 98.7 | 10.1 | 22.7 | 0.1 | 0.8 | 2.3 | 93.2 | 90.5 |
| | Clinic-REopt-B | 99.9 | 99.3 | 5.2 | 14.8 | 0.3 | 1.6 | 3.8 | 88.8 | 86.2 |
| | DT-H-REopt-A | 99.0 | 100.0 | 9.2 | 18.0 | 1.0 | 3.2 | 5.3 | 91.1 | 90.8 |
| | DT-L-REopt-A | 99.1 | 100.0 | 8.9 | 17.0 | 2.0 | 4.0 | 6.0 | 92.6 | 92.0 |
| P5 | DT-L-REopt-B | 99.5 | 99.9 | 9.3 | 17.7 | 1.9 | 3.8 | 5.7 | 91.2 | 90.6 |
| | Clinic-REopt-A | 99.5 | 99.9 | 12.0 | 22.3 | 2.1 | 4.5 | 6.8 | 91.8 | 91.0 |
| | Clinic-REopt-B | 99.5 | 100.0 | 12.2 | 22.4 | 2.0 | 4.2 | 6.2 | 90.2 | 89.3 |

Table E3: Dosimetric endpoint comparison for clinical and digital twin-based plans following online reoptimization (REopt) with fraction 2 CBCT (CB2). Underlined values indicate optimal plan quality scores per patient.

| | **CB2 REopt** | DIL (%) V100 | CTV (%) V100 | Bladder (cc) V20.8Gy | V14.6Gy | Rectum (cc) V23Gy | V17.6Gy | V13Gy | Urethra (%) D0.03cc | D10 |
|----|----------------|------|-------|-------|-------|------|------|------|-------|------|
| | DT-H-REopt-A | 98.5 | 99.9 | 8.4 | 16.3 | 1.0 | 3.4 | 6.6 | 92.1 | 90.4 |
| | DT-L-REopt-A | 97.0 | 99.8 | 8.2 | 16.6 | 1.5 | 4.1 | 7.8 | 94.6 | 91.7 |
| P1 | DT-L-REopt-B | 97.8 | 99.7 | 8.6 | 17.0 | 1.4 | 3.5 | 6.5 | 95.4 | 92.0 |
| | Clinic-REopt-A | 98.8 | 99.9 | 8.2 | 18.6 | 1.4 | 3.5 | 6.4 | 91.8 | 88.9 |
| | Clinic-REopt-B | 97.7 | 100.0 | 8.7 | 19.0 | 1.4 | 3.4 | 6.3 | 91.3 | 88.2 |
| | DT-H-REopt-A | 100.0 | 100.0 | 19.9 | 33.6 | 0.4 | 1.4 | 4.1 | 92.9 | 91.6 |
| | DT-L-REopt-A | 100.0 | 100.0 | 18.9 | 32.6 | 0.5 | 2.3 | 4.6 | 93.9 | 91.9 |
| P2 | DT-L-REopt-B | 100.0 | 99.9 | 18.3 | 31.8 | 0.5 | 2.4 | 4.9 | 93.8 | 92.0 |
| | Clinic-REopt-A | 100.0 | 99.8 | 20.6 | 34.1 | 0.4 | 1.9 | 3.9 | 93.8 | 91.0 |
| | Clinic-REopt-B | 100.0 | 99.8 | 19.6 | 33.5 | 0.5 | 2.1 | 4.3 | 93.8 | 91.1 |
| | DT-H-REopt-A | 99.9 | 99.8 | 9.5 | 16.8 | 0.9 | 2.4 | 4.0 | 100.6 | 93.5 |
| | DT-L-REopt-A | 99.7 | 99.9 | 9.6 | 17.2 | 0.9 | 2.3 | 3.7 | 93.6 | 98.9 |
| P3 | DT-L-REopt-B | 99.9 | 99.7 | 9.6 | 17.0 | 0.9 | 2.3 | 3.8 | 100.0 | 94.3 |
| | Clinic-REopt-A | 100.0 | 99.9 | 13.2 | 23.7 | 1.0 | 3.6 | 5.8 | 100.6 | 95.9 |
| | Clinic-REopt-B | 99.9 | 99.7 | 12.8 | 23.2 | 1.1 | 3.6 | 6.0 | 101.5 | 94.8 |
| | DT-H-REopt-A | 99.8 | 99.6 | 9.8 | 23.3 | 0.6 | 2.9 | 6.0 | 90.3 | 88.1 |
| | DT-L-REopt-A | 99.3 | 99.9 | 8.6 | 20.6 | 0.5 | 2.7 | 5.8 | 88.9 | 86.6 |
| P4 | DT-L-REopt-B | 99.0 | 99.4 | 8.6 | 20.2 | 0.6 | 2.9 | 5.9 | 87.4 | 85.1 |
| | Clinic-REopt-A | 100.0 | 100.0 | 9.2 | 24.1 | 0.7 | 3.2 | 6.3 | 95.1 | 92.6 |
| | Clinic-REopt-B | 99.9 | 100.0 | 9.3 | 22.7 | 0.8 | 3.1 | 6.2 | 89.1 | 87.5 |
| | DT-H-REopt-A | 99.6 | 100.0 | 9.9 | 19.7 | 1.0 | 2.6 | 4.6 | 90.6 | 89.1 |
| | DT-L-REopt-A | 99.6 | 100.0 | 10.0 | 19.8 | 1.6 | 3.7 | 5.8 | 91.0 | 90.2 |
| P5 | DT-L-REopt-B | 99.4 | 99.9 | 10.2 | 20.3 | 1.4 | 3.4 | 5.7 | 91.0 | 90.6 |
| | Clinic-REopt-A | 99.0 | 99.9 | 14.2 | 26.3 | 1.8 | 4.2 | 6.6 | 91.1 | 90.5 |
| | Clinic-REopt-B | 99.5 | 100.0 | 12.9 | 24.8 | 1.4 | 3.7 | 6.1 | 89.9 | 88.9 |

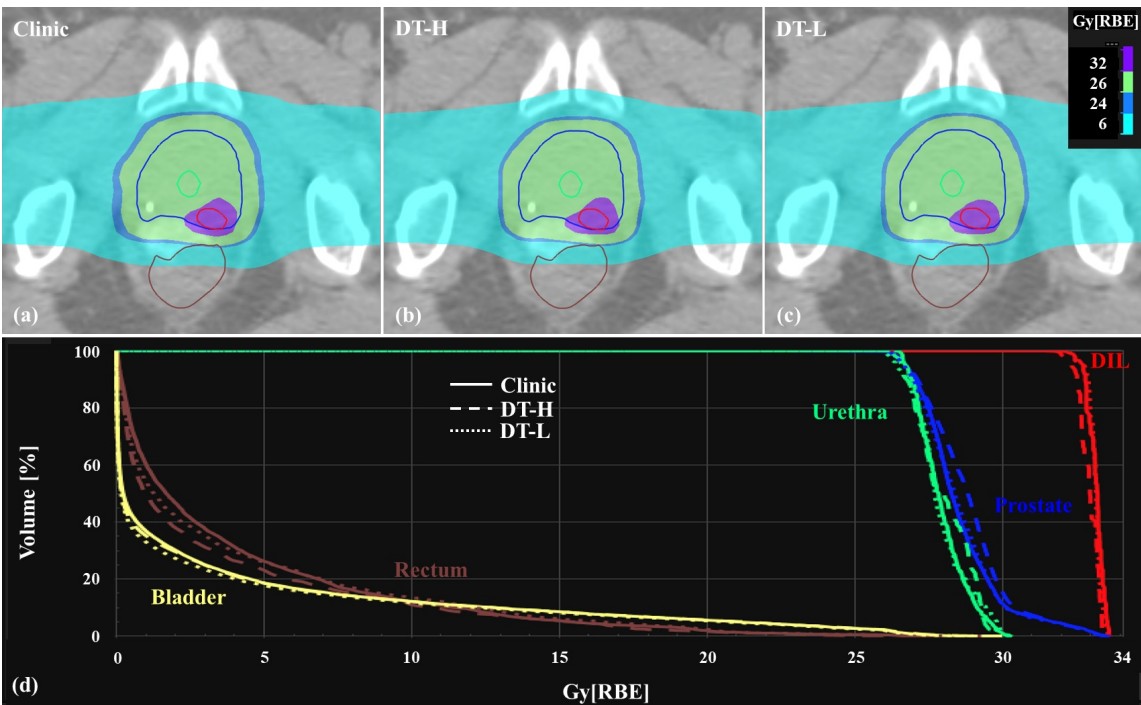

Figure E2: Dosimetric evaluation on daily CBCT for Patient 2 comparing clinical, DT-H, and DT-L reoptimized plans. DT-H and DT-L represent pCTs with maximum and minimum similarity to daily CBCT, respectively. Dose color wash for (a) clinical, (b) DT-H, and (c) DT-L plans with contoured structures: DIL (red), CTV (blue), bladder (yellow), urethra (green), and rectum (brown). (d) DVH analysis with solid, dashed, and dotted lines representing clinical, DT-H, and DT-L plans, respectively.

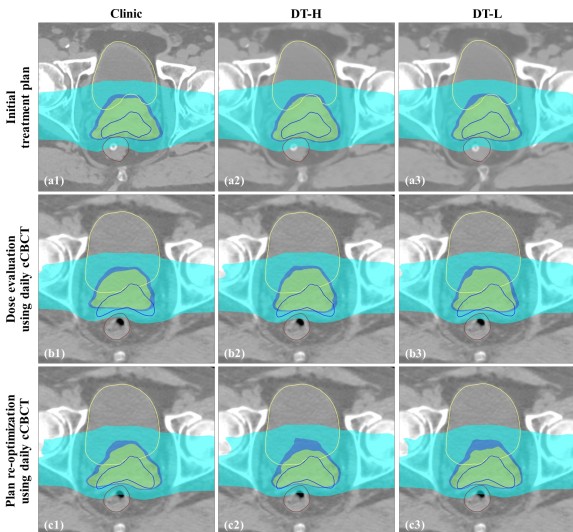

Figure E3: Dosimetric evaluation on daily CBCT for Patient 2 comparing clinical, DT-H, and DT-L reoptimized plans. DT-H and DT-L represent pCTs with maximum and minimum similarity to daily CBCT, respectively. Dose color wash for (a) clinical, (b) DT-H, and (c) DT-L plans with contoured structures: DIL (red), CTV (blue), bladder (yellow), urethra (green), and rectum (brown). (d) DVH analysis with solid, dashed, and dotted lines representing clinical, DT-H, and DT-L plans, respectively.

