# OpenReview forum: "A Deep Learning-Enabled Digital Twin Framework for Fast Online Adaptive Proton Therapy: A Validation Study in A Prostate SBRT Clinical Application"
_MIDL.io/2026/Conference — MIDL 2026 Poster_

### Official Review · Reviewer_xKEY · 2026-01-06

**Confidence:** 4
**Preliminary Rating:** 2
**Final Rating:** 4

**Summary:**

This paper presents a deep learning-based digital twin system designed to speed up online adaptive proton therapy for prostate cancer. It employs a VoxelMorph-based multi-atlas deformable image registration process to pre-generate a library of predicted CT images for a new patient by adjusting previous patient images. When daily imaging occurs (CBCT), the system selects or quickly re-optimizes a treatment plan from this library, guided by a knowledge-based composite score adapted from the ProKnow system. This score evaluates target coverage and organ-at-risk sparing. In a retrospective study involving 43 prostate SBRT patients, the framework achieved an average re-planning time of approximately 5.5 minutes, compared to 19.8 minutes in conventional clinical workflows, representing a 72% reduction while maintaining comparable plan quality. The authors find that the digital twin plans resulted in similar or slightly better dosimetric outcomes (composite plan quality score approximately 157 versus 154 for clinical plans) and satisfied clinical dose constraints, with DIL (dominant intraprostatic lesion) coverage around 99.5% and similar bladder and rectum sparing as the original plans. Overall, this paper validates the integration of deep learning-based registration with knowledge-based evaluation to facilitate near-real-time adaptive proton therapy in a prostate SBRT context.

**Strengths:**

- Addresses a relevant clinical problem: The paper tackles the significant issue of speeding up online adaptive radiotherapy. Faster re-planning is vital for hypofractionated treatments like prostate SBRT. The authors show a notable reduction in plan adaptation time (5.5±2.7 minutes versus 19.8±11.9 minutes). This improvement—72% faster than current workflows—represents a meaningful practical contribution.

- Maintained plan quality: Despite the acceleration, the adapted plans preserved equivalent dosimetric quality. The composite plan scores, which are based on clinical criteria, were slightly higher for the digital twin plans (157.2±5.6 versus 153.8±6.0 for clinical plans). Critical metrics, like DIL V100 (~99.5%), CTV V100 (~99.8%), and OAR doses, stayed within clinical limits. This indicates that the framework did not compromise treatment quality for speed.

- Comprehensive evaluation: The study provides a thorough validation on a reasonably sized dataset (43 patients, 210 CBCTs). The authors include detailed dosimetric analysis for targets and various OARs (bladder, rectum, urethra) to ensure that the adapted plans meet rigorous SBRT constraints. By using an established knowledge-based scoring system (ProKnow) with nine clinically motivated metrics, they aimed to quantitatively assess plan quality in a single composite measure, enabling systematic plan comparison.

- Integration of deep learning in workflow: Using a VoxelMorph-based deformable registration to create a patient-specific atlas of anatomical variations is a sensible application of deep learning for this issue. The unsupervised VoxelMorph model delivers fast deformable image registration, which is essential for efficiently generating numerous predicted CT images. The pipeline cleverly utilizes population data (motion patterns from previous patients) to anticipate a new patient's anatomical changes, aligning with the “digital twin” concept. This combination of learning with domain knowledge (atlas of past cases plus knowledge-based planning) is a strength that reflects an understanding of both machine learning and clinical context.

**Weaknesses:**

- Limited novelty of methodology: The paper's technical contributions are incremental. It primarily uses standard components, such as a conventional VoxelMorph U-Net model for deformable registration, without clear innovations in the learning algorithm or registration method. No new network architecture or learning approach is introduced; the deep model is directly derived from previous work (Balakrishnan et al., 2019) and applied to this dataset. The pipeline (multi-atlas deformable registration, treatment plan library, and knowledge-based evaluation) resembles a specific integration of known techniques rather than representing a novel methodological advance for the medical imaging field. This lack of a clear innovative contribution raises questions about whether the work offers new insights beyond an effective case study.

- Heuristic atlas selection strategy: The multi-atlas framework depends on hand-tuned heuristics that are not rigorously justified. The authors select the “top 20%” of patients from the database as atlases based on a composite image similarity score that combines SSIM, NCC, and LPIPS equally. This 20% threshold (about 8–9 patients out of 43) and the equal weighting of similarity metrics are arbitrary choices; the paper does not provide analysis to support them. It is unclear if 20% is the best choice, or if altering the number of atlases or metric weightings would change the results. The atlas selection thus feels ad-hoc, and the robustness of this choice lacks demonstration—a potential methodological weakness.

- Reliance on a composite score without validation: Plan quality is assessed using a ProKnow-based composite metric that combines nine dosimetric criteria. Although this is convenient for automation, the paper does not verify whether optimizing this score truly leads to the best clinical outcomes. There is no analysis to show whether the composite scoring is necessary compared to simpler selection criteria. For example, would selecting the plan that maximizes target coverage and meets strict OAR constraints yield a different outcome than the ProKnow score? The approach assumes that the weighted sum of scores adequately captures plan quality, but without evidence, this could hide trade-offs, as a plan might score highly overall despite a minor OAR violation. The lack of analysis on the impact of the scoring system means we must take it on faith that the chosen metric is suitable—a risky proposition if the composite score has not been independently validated for this use case.

- No external or independent validation: All results are based on a single institutional dataset (patients from one center with one scanner and protocol). There is no external validation with data from another hospital or a different patient group to prove generalizability. The framework was tuned and tested on the same pool of 43 patients, likely through cross-validation, though details are not clearly outlined. This raises concerns that the system may be particularly tailored to that institution’s imaging and patient characteristics. The authors also acknowledge that using a single-institution cohort “potentially limits generalizability.” Without external validation, it is unclear if the claimed performance would hold true in different clinical settings or with patients whose anatomies aren’t represented in the training data.

- Evaluation missing key baselines and analyses: The study does not compare to other potential adaptation methods beyond the current clinical workflow. For instance, it lacks a comparison to a simpler atlas-based approach without deep learning, such as using a standard DIR algorithm or fewer atlas cases. This gap makes it difficult to isolate the advantages of the learning-based component. Similarly, the authors did not provide any direct registration accuracy metrics like contour alignment or Dice scores for anatomical structures; they only report image-similarity indices. Without evaluating how well critical structures (prostate, bladder, etc.) are aligned by VoxelMorph, it’s challenging to determine if the registration quality is adequate for high-precision dose adaptation. Important aspects like the accuracy of propagated contours or dose distribution variations on the deformed CTs are not discussed. These gaps weaken the evidence for the framework’s effectiveness and safety.

- Potential overclaim and scope limitations: The paper suggests that the work enables “real-time” adaptive therapy, but a re-planning time of about 5–6 minutes (plus imaging and QA overhead) does not precisely qualify as real-time. It is an improvement, but calling it "real time" is a bit optimistic when it still requires minutes (the authors themselves note it “suggests… within 10 minutes” feasibility). Furthermore, the solution is domain-specific, focusing on prostate SBRT with a particular protocol (DIL boost). Adapting this pipeline to other sites or treatment plans would require additional development, such as different atlases and retraining VoxelMorph for different anatomies. Thus, the impact on the broader medical imaging field may be limited; it serves more as a tailored clinical system validation than a general methodological advance. In summary, the submission’s contributions, while valuable as proof-of-concept for one application, lack the novelty and broad applicability expected for a full paper at MIDL.

**Detailed Comments:**

- VoxelMorph usage without innovation: The authors develop their deformable image registration module using the basic VoxelMorph framework, which is a well-known unsupervised learning method. They do not discuss any recent advancements, such as VoxelMorph++, transformer-based registrars, or segmentation guidance, that could improve accuracy. Essentially, the paper depends on a 2019 model and does not propose any improvements. This approach is acceptable for a validation study, but it seems technically derivative for a research contribution. The paper would be more robust if it either introduced an improved registration method or at least compared the chosen VoxelMorph model with other registration techniques. For instance, comparing it against a conventional DIR or a newer DL model would help justify that VoxelMorph was the best choice. As it stands, it is unclear if the modest image similarity improvements reported, such as SSIM 0.87 versus 0.82 over planning CT, result from the multi-atlas approach alone or could be obtained through simpler methods.

- Multi-atlas strategy and lack of justification: The concept of using population motion data to generate possible anatomies beforehand is intuitive and has been explored in previous adaptive therapy research, which the authors cite. However, the specific implementation here raises questions. The composite similarity metric (SSIM+NCC+LPIPS) used to select similar patients is presented without justification for those components or their equal weighting. LPIPS is an interesting addition; it is a learned perceptual metric often used in computer vision. Its application for CT similarity is novel but not explained. Did they find it correlates better with anatomical accuracy? An ablation study showing the effects of including or excluding LPIPS, or using just one metric, would have been informative. Additionally, choosing the top 20% of patients as atlases seems arbitrary. Perhaps this fraction was adjusted based on the dataset, but if so, that is not reported, which raises concerns about potential overfitting. What if a new patient has a somewhat unique anatomy? Limiting to 8 or 9 atlases might miss relevant variations. On the other hand, using too many atlases could introduce unnecessary or dissimilar anatomy. The paper would benefit from a sensitivity analysis on atlas count or threshold, or at least a rationale for the choice, such as confirming that “20% produced the best results on a validation set.” Without this, the multi-atlas approach appears as a fixed rule with uncertain reliability.

- Evaluation of the knowledge-based planning metric: The authors use the ProKnow scoring system to evaluate plan quality, combining nine sub-scores into one composite score. While this creates a single number to rank plans, it also obscures the multi-objective nature of radiotherapy planning. The paper does not present any scenario where the composite score might be misleading. For example, did all plans that scored higher actually meet all clinical constraints? The text mentions a case where the bladder V20.8Gy in the digital twin plan slightly exceeded the nominal constraint (11.4±4.2 cm³ versus <10 cm³), yet the composite score still called it “optimal.” This suggests that the scoring function might permit minor violations if other metrics score well, since it is a weighted sum. A clinician might accept this trade-off or not. The authors should clarify how constraint satisfaction is enforced or reflected in the score. An ablation study could involve taking the same scenarios and selecting plans by a more straightforward rule, like the lowest DIL coverage deviation or a strict satisfaction of constraints, to see if using the composite score leads to different results. Without such analysis, the use of ProKnow, despite its grounding in prior work, feels like a hidden decision-maker in the process. The paper would be more convincing if it showed that this knowledge-based score aligns with expert judgment or outcomes.

- Reproducibility and clarity: As a reviewer, I found some details underspecified, which could limit reproducibility. It is not clear how the VoxelMorph model was trained and validated. Did they use data from all 43 patients to train the network and then perform a leave-one-out test for each “new” patient simulation? Or was there a split, such as training on 38 patients and testing on 5? The text suggests the framework was developed and validated on the same 43 patients, implying a possible cross-validation approach, but this should be clearly explained. There is also a confusing inconsistency regarding the fractionation: the dataset description states five-fraction SBRT for all patients, but the scoring functions and protocol are based on a two-fraction SBRT regimen (the 2SMART trial). This inconsistency is not addressed—did the patients actually receive 5 fractions, and if so, why is a 2-fraction dose schedule used for evaluation? Or did 5 patients receive a DIL boost in 2 fractions while others received it in 5 fractions? This needs clarification. Such ambiguities make it difficult to fully trust and reproduce the experimental setup. Additionally, no information is provided about how long the offline pre-computation took, such as registering 26,312 images and optimizing many plans. If that process is time-consuming or requires significant manual work, it could limit practical use, but the paper does not address these engineering details.

- Scope of validation: The results are positive but specific to one disease site and modality. The authors correctly point out that intrafraction motion is not modeled—in proton SBRT for prostate treatment, intrafraction motion, such as due to bowel gas or patient movement during beam delivery, can be significant over several minutes. Ignoring this means the “digital twin” might not fully represent reality for longer treatment sessions. In a true online adaptation setting, one would also need to consider the extra imaging dose and time required for daily CBCT—the study does not discuss this practical aspect since it is retrospective. Furthermore, applying this approach to more anatomically complex sites, such as lung tumors affected by respiration or head-and-neck regions with significant anatomical differences, would be challenging. The current pipeline might not transfer directly to these complexities. The paper shows feasibility in a favorable case, with relatively rigid pelvic anatomy and two-field proton plans, but does not convincingly demonstrate that the framework would easily generalize. This somewhat limits the scientific impact of the work, as it may be seen as a one-off solution for a specific clinical scenario rather than a broadly applicable advance.

- Comparison to prior work: The authors reference a few studies on adaptive therapy and deep learning registration but could position their work more effectively. For instance, how does this “digital twin” differ from other adaptive planning methods or recent studies on “dose prediction” or “plan transfer”? There have been earlier attempts at atlas-based adaptive planning and using statistical motion models for adaptive radiotherapy. The paper does not clearly differentiate itself from those efforts beyond the term “digital twin.” It would strengthen the paper if the authors explained what is genuinely new in this approach. Is it the first to use a deep learning DIR for plan library generation? Possibly, but they do not clearly state this or provide a comparison with a traditional DIR method. Without demonstrating a significant improvement in either speed or accuracy due to the deep learning, it is hard to regard the method as a major research contribution.

- In summary, the submission appears to be a system validation paper. It assembles known components, such as deep deformable registration, multi-atlas modeling, and knowledge-based evaluation, into a pipeline and tests it on a clinical dataset. This is commendable from an engineering perspective and relevant for clinical implementation, but it lacks significant novelty or thorough analysis of the method's unique aspects from a research standpoint. The results are promising, yet the paper stops short of offering new methodological insights or in-depth evaluations that would elevate it beyond a case study. I encourage the authors to address the above concerns to enhance the paper's contribution to the community.

**Justification Of Final Rating:**

The authors' rebuttal has updated my understanding of the submission's main contributions. I now believe that the study exhibits appropriate characterization as a system-level, clinically based feasibility/validation, rather than as an advancement in algorithm development, for deformable registration. The rebuttal has additionally clarified several concerns of mine: (i) There is more clear distinction between the offline "digital-twin" instantiation and the online, on-couch adaptation. The ~5.5 minutes reported for the "adaptation" time corresponds entirely to the clinically relevant online phase (i.e., optimizing and calculating the dose); (ii) The discrepancy regarding the greater fractional number of fractions for the clinical data and fewer fractions used for feasibility demonstrating includes both a five-fraction clinical evaluation while demonstrating feasibility under a two-fraction limit; (iii) "Unseen data" has been established through a leave-one-patient-out, patient/population informed instantiation approach; and (iv) The intended use of the "ProKnow composite score" reflects physician-devised/useful and transparent support for decision making post "hard acceptance". Although I still view the authors’ methodological innovation is limited (I would welcome additional ablation studies using atlas-threshold sensitivity and alternative DIR baseline methods as well as additional explicit metrics for registration/contour accuracy) and the clarified workflow rationale, safety-based decision logic and time/quality trade-off in actual clinical conditions lend credibility to the work and support a relevant contribution to MIDL's target audience of translational research. For these reasons I will vote to "accept weakly."

**Justification Of The Preliminary Rating:**

At this stage, I am leaning towards "weak reject" for this submission. On one hand, the paper targets an important problem of timely relevance in adaptive radiotherapy and demonstrates a solution that works on the cases tested - the system achieves notable time reduction with maintained plan quality, which is not a trivial achievement and well-founded on solid results. The manuscript is well-written in parts. These positive aspects would therefore suggest the work has merit and could eventually have clinical benefit.

Conversely, being a conference paper on MIDL, innovation on the methods is quite low. The work is all about reusing available methods (Deep Registration, Atlas-Based Planning, and Knowledge-Based Scoring) without any new algorithms and/or concepts that push the boundaries on medical imaging or learning algorithms. The work is claimed to be a “validation study” by the writers, and truly, it is so because it is very much valid for systems-level work but may be more on the innovation-neutrality side regarding its scientific value. The fact that it has no external validation and there are many unanswered questions on its design (pointed out above) adds to the suspicions on its scientific merit. Based on these, the work, in its form and content, does not make any imperative on having a clear accept status. It is on the borderline category where it would take extra work to convince that it is no longer a special treatment on what is already available.

**Questions To Address In The Rebuttal:**

- What is the new contribution of using VoxelMorph here? Did the authors consider or compare it to other registration methods, including traditional DIR or newer DL models? How does VoxelMorph specifically improve performance, and does it offer better accuracy or speed than those alternatives? Essentially, why is this considered a learning-based innovation rather than just using an existing tool?

- Why choose the top 20% atlases and use equal-weight SSIM, NCC, and LPIPS? Is there any evidence that this choice is optimal? For example, did the authors test the top 10%, 20%, and 50% or try applying different weights to the similarity metrics? More results or justification here would help alleviate concerns that the atlas selection is arbitrary or overfit.

- How does the ProKnow composite score affect decision-making? Can the authors provide insight or data on whether this composite scoring actually changes which plan is chosen compared to using simple constraint satisfaction or target coverage criteria? If possible, provide an example: did the highest-scoring plan ever differ from one that maximized DIL coverage while meeting all hard constraints? Essentially, validate that the composite metric is appropriate and that no good plans are overlooked or mis-ranked by this method.

- Generalizability and validation: Have the authors tested the framework on any patients or data not included in the original 43-patient database, such as a true hold-out test or an external dataset? If not, how do they address concerns about performance on unseen data? A clear statement on how the data were split for training versus testing, such as whether it involved leave-one-patient-out cross-validation, and whether any external cases were tested would help establish credibility.

- Clarification of protocol and dataset: The paper mentions five-fraction treatments in the dataset but uses a two-fraction SBRT dose schedule for plan evaluation. Please clarify this discrepancy. Were the 43 patients treated with five fractions each, as stated? If so, why are the plan quality scoring functions taken from a two-fraction trial? This confusion needs to be resolved for a proper understanding of the results.

- Reoptimization timing and workflow: The reported 5.5-minute reoptimization is impressive. What components does this include? Does it account for image registration time and dose calculation, or is it mainly the optimization of precomputed plans? If VoxelMorph registration is essentially instantaneous, taking just a few seconds, one might wonder if a faster conventional method could achieve similar online speed. A breakdown of the timing, including imaging, contour propagation, plan selection, and dose calculation, would help clarify the source of time savings and ensure there are no hidden delays, such as data transfer or user intervention, that are not accounted for. This will also emphasize the role of the deep learning model in the time savings.

---

> ### Author Response · Authors · 2026-01-21
> **Summary by the reviewer**
>
> Thank you very much for the general summary and suggestion of our manuscript. We very much appreciate the reviewer for contributing expertise to our manuscript. We have made a thorough revision of the manuscript to clarify the technical details and improve the manuscript structure based on the reviewer’s comments. The primary innovation of this work is the development of a novel framework that directly addresses a critical clinical challenge in online adaptive radiotherapy, which remains a major limitation in current radiotherapy practice. Beyond the methodological contribution, this study also introduces a significant clinical innovation, as it is, to our knowledge, the first work to apply a digital twin framework to address real-time decision-making challenges in online adaptive radiotherapy.

---

> ### Author Response · Authors · 2026-01-21
> **Detailed Comment 1.**
>
> We appreciate the reviewer’s detailed and thoughtful critique and agree that this work does not propose a new deformable image registration architecture. This is intentional and reflects the scope of the contribution. The primary objective of this study is to develop, operationalize, and validate a digital twin framework for online adaptive proton radiotherapy, rather than to advance the state of the art in deformable image registration algorithms. In this context, the learning-based registration component serves as an enabling module within a larger system that addresses a long-standing translational challenge in proton therapy: generating clinically beneficial adaptive treatment plans within a strict on-couch time window. Unlike conventional photon radiotherapy, proton therapy planning involves substantially more complex physics, including range uncertainty, Bragg peak sensitivity, tissue heterogeneity, and multiple Coulomb scattering. As a result, there is currently no mature, fully automated online adaptive planning solution for proton therapy comparable to those available for photon therapy. The critical bottleneck is therefore not marginal improvements in image similarity metrics, but rather system-level integration that enables fast, reliable, and dosimetrically safe adaptive plan generation under real clinical constraints.
>
> Within the proposed framework, deformable image registration is performed prior to the first treatment fraction as part of digital twin instantiation. Consequently, registration runtime and marginal accuracy gains have limited impact on treatment-day efficiency, which is dominated by adaptive plan evaluation and selection while the patient is on the treatment couch. The performance of the framework is therefore not bounded by the choice of VoxelMorph itself. Instead, it is governed by whether the overall system can deliver adaptive proton SBRT plans within approximately 10 minutes that meet clinical dosimetric requirements and avoid excessive normal tissue dose. VoxelMorph was selected because it is a well-validated, robust, and unsupervised registration model that is sufficiently accurate for population-based inter-patient deformation in pelvic anatomy. Importantly, this study demonstrates that such a model, when embedded in a multi-atlas digital twin framework, can achieve dosimetrically acceptable proton dose calculations, which is a far more stringent requirement than image similarity alone. This result is nontrivial, given that proton dosimetry is substantially less forgiving than photon therapy with respect to CT-based material characterization errors.
>
> We agree that comparing against newer registration architectures or conventional DIR methods could be informative for algorithm benchmarking. However, such comparisons would shift the focus of the paper toward registration performance optimization rather than toward the system-level feasibility and clinical deployment of a digital twin for proton therapy, which is the central contribution of this work. As a system validation study, our emphasis is on workflow design, offline and online separation, decision logic, and clinical plan quality rather than on incremental algorithmic improvements. We will revise the manuscript to include this in Discussion to make this positioning more explicit, clarify that the contribution is system-level rather than algorithmic, and avoid overstating novelty in the registration component. We will also note that the framework is agnostic to the specific registration backbone and can readily incorporate future advances, such as transformer-based or segmentation-guided registrars, without altering the core digital twin architecture.

---

> ### Author Response · Authors · 2026-01-21
> **Detailed Comment 2.**
>
> We appreciate the reviewer’s detailed comments regarding the multi-atlas strategy, the choice of similarity metrics, and the atlas selection threshold, and we agree that these design choices warrant careful explanation. The multi-atlas selection step in the proposed framework is not intended to define final plan quality or to identify an optimal anatomical match. Instead, it serves as a computationally efficient preselection mechanism that reduces the number of candidate anatomies for which full adaptive proton treatment plans must be generated. Because adaptive proton SBRT planning is computationally expensive, particularly when Monte Carlo dose calculation is required, this preselection step is necessary to ensure that the overall workflow remains clinically feasible within a strict on-couch time window. Importantly, image similarity is used only to filter candidates. Final plan selection is governed exclusively by dosimetric performance, including target coverage, organ-at-risk sparing, and compliance with all clinical constraints. As a result, the framework's ultimate performance is determined by plan quality rather than by image-similarity metrics.
>
> The choice of SSIM, NCC, and LPIPS with equal weighting reflects a conservative, complementary strategy rather than an optimized image-matching rule. SSIM captures local structural similarity, NCC reflects global intensity agreement, and LPIPS provides a learned perceptual measure sensitive to texture-level differences that may influence contour propagation and material characterization. While LPIPS has been primarily used in computer vision, its inclusion here is motivated by the need to capture subtle anatomical texture differences that are not well described by intensity or structural metrics alone. We do not claim that LPIPS is optimal or uniquely superior for CT similarity, nor do we assert that it directly correlates with registration accuracy or dosimetric outcomes. Rather, it is used as part of a balanced metric set to avoid reliance on a single similarity criterion.
>
> The selection of the top 20 percent of atlases should not be interpreted as a tuned or optimal threshold. This value was chosen to balance two competing system-level requirements: retaining sufficient anatomical diversity to capture population variability while limiting the number of candidate plans to maintain computational tractability. We did not perform a formal sensitivity analysis over atlas counts or metric weights, such as comparing 10 percent, 20 percent, or 50 percent thresholds, because such analysis would shift the focus of the study toward heuristic parameter tuning rather than toward the primary goal of evaluating system-level feasibility, workflow integration, and clinical plan quality. To mitigate concerns about bias or overfitting, we explicitly examined candidates with both high and low similarity scores within the selected subset and confirmed that dosimetric trends behaved as expected. Regarding anatomically unique patients, the framework does not rely on a single atlas or a narrow representation. Even within the top 20 percent subset, multiple diverse candidate anatomies are retained, and all candidate plans must independently satisfy hard clinical constraints. If no acceptable plan can be generated, the workflow transitions to a standard clinical replan protocol using a new planning CT. This safeguard ensures that unusual anatomy does not compromise patient safety.
>
> In summary, the multi-atlas strategy is designed as a robust, computationally practical filtering step, not as a fixed or optimized image-similarity rule. The innovation of this work lies in integrating population-informed atlas selection, learning-based registration, and clinically grounded dosimetric decision-making into a digital twin framework for adaptive proton SBRT. The reliability of the framework is ultimately validated by its ability to generate clinically acceptable plans within a bounded time window, rather than solely by image similarity performance.

---

> ### Author Response · Authors · 2026-01-21
> **Detailed Comment 3.**
>
> We thank the reviewer for the thoughtful comment regarding the use of a composite ProKnow score and the concern that a single scalar metric may obscure the inherently multi-objective nature of radiotherapy planning. We fully agree that radiotherapy plan selection cannot be reduced to a purely numerical optimization problem. In clinical practice, treatment decisions are guided by a combination of evidence from clinical trials, physician experience, and patient-specific considerations. In particular, strict satisfaction of all organ-at-risk (OAR) constraints may not always be achievable in challenging anatomical or adaptive scenarios, and clinically acceptable plans often involve deliberate trade-offs to maintain tumor control while minimizing toxicity.
>
> ProKnow is a clinically established, transparent, rule-based plan evaluation system, not an optimization engine or a machine learning model. Its composite score is derived from multiple predefined dose–volume objectives for targets and OARs, explicitly configured and approved by treating physicians in accordance with institutional clinical priorities. The scoring functions and weights are therefore not hidden decision variables, but rather an explicit encoding of physician-defined treatment goals and acceptable trade-offs. In our framework, ProKnow is not used to replace clinical constraints. All candidate plans are first required to satisfy hard clinical acceptability criteria. Among clinically acceptable plans, the composite score is used to rank overall plan quality holistically. This reflects routine clinical decision-making, where the plan with the highest target coverage is not necessarily preferred if it leads to excessive normal tissue dose. Indeed, a substantial fraction of adaptive replanning in practice is driven by OAR dose escalation rather than target under-coverage.
>
> The reviewer correctly notes the example where the bladder V20.8 Gy in one digital-twin–selected plan slightly exceeded the nominal constraint while still achieving a high composite score. This behavior is intentional and clinically realistic: the scoring system allows minor, physician-defined deviations to be offset by meaningful improvements in other clinically relevant metrics. Whether such a trade-off is acceptable ultimately rests with the clinician, and ProKnow is designed to reflect this judgment rather than enforce rigid binary pass/fail rules. We agree that alternative selection rules (e.g., strict constraint satisfaction or prioritizing dominant intraprostatic lesion coverage alone) could yield different plan choices. However, our goal was not to claim that the composite score is universally optimal, but to demonstrate that a knowledge-based, physician-configured evaluation framework can be integrated into a digital twin workflow to support clinically meaningful plan selection. Importantly, in multiple cases, we observed that plans with slightly lower target coverage, but substantially improved OAR sparing, were favored, consistent with real-world clinical reasoning.
>
> We have clarified this rationale in Discussion of the revised manuscript and emphasized that the composite ProKnow score serves as a structured decision-support layer rather than a hidden decision-maker. Its role is to embed expert clinical priorities into automated plan selection while preserving transparency and physician oversight, thereby aligning the digital twin framework with actual clinical practice rather than with isolated dosimetric objectives alone.

---

> ### Author Response · Authors · 2026-01-21
> **Detailed Comment 4.**
>
> We thank the reviewer for pointing out areas where additional clarification is needed to ensure reproducibility. We have revised the manuscript to explicitly describe the training and validation strategy of the VoxelMorph-based deformable image registration (DIR) component. The framework employs a leave-one-patient-out strategy at the system level. For each evaluated patient, that patient is treated as unseen data relative to the population-based deformation priors. Specifically, for a given “current” patient, the DIR model is trained exclusively using imaging data from the remaining 42 patients as moving images, while the current patient’s planning CT is used only as the fixed reference during deployment. No longitudinal images from the current patient are used for training. This design avoids information leakage while mirroring the intended clinical workflow, in which a digital twin is instantiated for a new patient using population-informed priors.
>
> We emphasize that the learning-based DIR module is not intended to benchmark generic registration performance, but to encode population anatomical priors that enable synchronization within the digital twin framework. As such, every evaluated case represents unseen data at the system level. While we agree that external validation is an important future direction, the current study intentionally focuses on single-institution system validation. For proton therapy, meaningful multi-institutional validation requires harmonized CT-to-material calibration, as proton dose calculations are highly sensitive to scanner-dependent stopping-power conversion. Without such harmonization, external data may introduce confounding dosimetric errors unrelated to the proposed framework. We have clarified this staged validation rationale in the revised manuscript.
>
> We also thank the reviewer for noting the apparent inconsistency in fractionation. This study is retrospective, and all imaging data were collected from 43 prostate cancer patients treated at our institution under a standard five-fraction SBRT protocol. All planning CT and daily CBCT images were acquired during routine clinical care. To demonstrate feasibility, adaptive plans were evaluated using dosimetric constraints derived from a two-fraction prostate SBRT clinical trial. This choice was intentional and does not imply that patients were treated with two fractions. A two-fraction setting represents a stringent evaluation scenario with higher dose per fraction and tighter organ-at-risk constraints, allowing us to test whether the digital twin framework can generate clinically acceptable adaptive plans under demanding conditions. The fractionation used for evaluation defines the clinical criteria, while the underlying anatomy and imaging data remain unchanged. We have revised the manuscript to clearly distinguish between the treatment protocol used for data collection and the evaluation protocol used for plan assessment.
>
> Finally, regarding computational feasibility, the framework is designed with a clear separation between offline pre-computation and online deployment. The offline phase includes training population-based DIR models, synchronizing prior CBCTs, and optimizing candidate plans, which involves registering 26,312 images and is computationally intensive but fully automated and performed once per institutional database. The online phase, which is clinically relevant, involves instantiating a patient-specific digital twin using precomputed population priors and can be executed within clinically acceptable time constraints. We have added these engineering details to the revised manuscript to improve transparency regarding practical deployment.

---

> ### Author Response · Authors · 2026-01-21
> **Detailed Comment 5.**
>
> We thank the reviewer for the thoughtful assessment of the scope and limitations of the proposed framework. We agree that the current study focuses on a single disease site and modality, and we have clarified in the revised manuscript that this work is intended as a feasibility and system-level validation rather than a demonstration of universal generalizability. The primary objective of this study is to address inter-fractional anatomical variation, which represents a major driver of adaptive replanning in prostate proton SBRT. The proposed digital twin framework is designed to proactively model and evaluate population-informed inter-fractional scenarios using routinely acquired daily CBCT. We explicitly acknowledge that intrafraction motion, such as bowel gas changes or patient movement during beam delivery, is not modeled in the current implementation. This limitation is intentional and reflects both the scope of the problem addressed and the available imaging information. Modeling intrafraction motion would require additional instrumentation, such as fluoroscopic or time-resolved kV imaging, which cannot be inferred reliably from CBCT alone. Incorporating such data sources to extend the digital twin toward intrafractional dynamics is an important direction for future work.
>
> We also agree that the study does not address practical considerations related to additional imaging dose or acquisition time in a true online adaptive setting. Because this work is retrospective, no additional imaging beyond standard-of-care CBCT was required. The current framework leverages imaging that is already routinely acquired in prostate SBRT workflows, and therefore does not introduce incremental imaging burden in its present form. We have clarified this point and note that prospective deployment would require careful consideration of imaging frequency, dose, and integration with clinical workflow. Regarding generalization to more anatomically complex sites such as lung or head-and-neck cancers, we agree that these scenarios pose additional challenges, including respiratory motion, greater soft-tissue deformation, and more complex beam arrangements. We do not claim that the current instantiation would directly transfer without modification. However, the digital twin framework itself is modular and disease-agnostic: its components, population-informed deformation modeling, scenario generation, and knowledge-based plan evaluation, are not inherently restricted to pelvic anatomy or two-field proton plans. Rather, the present study represents a necessary first step in a favorable and clinically controlled setting, consistent with how clinical digital twins are typically developed and validated.
>
> In summary, this work demonstrates the feasibility and clinical relevance of prostate proton SBRT, a setting in which inter-fractional anatomical variation is both common and clinically impactful. We have revised the manuscript to more explicitly frame the contribution as a system-level demonstration, clearly delineate the boundaries of applicability, and outline intrafraction modeling, additional imaging integration, and the extension to more complex disease sites as future directions.

---

> ### Author Response · Authors · 2026-01-21
> **Detailed Comment 6.**
>
> We thank the reviewer for encouraging a clearer positioning of this work relative to prior adaptive radiotherapy and learning-based planning studies. We agree that the novelty of the proposed approach lies not in a single algorithmic component, but in the system-level integration and clinical feasibility of a digital twin (DT) framework for adaptive proton SBRT, and we have revised the manuscript to make this distinction explicit. Prior work on adaptive planning, including atlas-based approaches, statistical motion models, dose prediction, and plan transfer methods, typically focuses on one isolated task, such as predicting dose distributions, transferring fluence maps, or deforming structures. In contrast, the proposed DT framework is designed to proactively model a patient’s future treatment course by synchronizing population-based anatomical trajectories with the current patient and embedding this information into a clinically usable adaptive planning workflow. The goal is not to predict dose from images alone, but to enable rapid generation and evaluation of fully physics-based treatment plans that satisfy clinical constraints and are acceptable to physicians.
>
> While deep learning–based DIR has been explored previously, its role here is fundamentally different. The DIR module is not used as an end in itself, nor is it optimized for benchmark registration accuracy. Instead, it serves as an enabling component that allows prior patients’ CBCTs to be synchronized into the current patient’s anatomical space, forming the foundation of a patient-specific digital twin. This population-informed synchronization enables offline precomputation of a large plan library and supports near-real-time adaptive plan selection. To the best of our knowledge, no prior proton SBRT adaptive framework has demonstrated clinically acceptable plan generation and selection within approximately 10 minutes using fully physics-based optimization, without online reoptimization. We intentionally evaluate the framework using system-level metrics, including plan quality, organ-at-risk sparing, and time to generate a deliverable adaptive plan, rather than isolating the performance of individual modules such as DIR accuracy. This choice reflects clinical reality: at our institution, adaptive plans are not accepted based solely on image- or dose-prediction fidelity. Physicians and physicists require transparent, physics-constrained plans that can be interrogated and justified, which limits the clinical adoption of purely data-driven dose prediction or black-box plan transfer methods, despite their computational appeal.
>
> Finally, we acknowledge that a direct comparison with traditional DIR or atlas-based approaches could further strengthen the work, and we have clarified in the revised manuscript that the primary contribution is demonstrating a feasible, end-to-end digital twin–driven adaptive proton therapy workflow, rather than claiming superiority of deep learning DIR in isolation. We believe this system-level demonstration addresses a critical gap in current adaptive proton SBRT practice and provides a foundation for future methodological comparisons and extensions.

---

> ### Author Response · Authors · 2026-01-21
> **Detailed Comment 7.**
>
> We thank the reviewer for the thoughtful summary and for recognizing the engineering rigor and clinical relevance of the proposed system. We agree that this work is best characterized as a system-level validation study, and we have revised the manuscript to explicitly frame it as such. This positioning is intentional rather than a limitation: the primary contribution of this work is not a new standalone algorithm, but the first end-to-end demonstration of a clinically feasible digital twin framework for adaptive proton SBRT, evaluated under realistic clinical constraints.
>
> While the individual components—deep deformable registration, multi-atlas modeling, and knowledge-based plan evaluation—have been studied previously in isolation, their integration into a coherent, deployable digital twin pipeline is non-trivial and has not been previously demonstrated for proton SBRT. In particular, this work shows how population-informed anatomical synchronization, offline scenario precomputation, and physics-based plan selection can be combined to enable adaptive plan generation within clinically actionable time frames, without relying on black-box dose prediction or online reoptimization. To the best of our knowledge, no prior study has demonstrated adaptive proton SBRT plan generation and selection at this system level within approximately 10 minutes using fully physics-constrained planning. We respectfully argue that the scientific contribution lies in identifying and validating what level of modeling, learning, and computation is sufficient to make digital twin–driven adaptive proton therapy clinically viable today. Rather than optimizing individual modules in isolation, we focus on system-level questions that are underexplored in the literature: how to define “unseen data” in a digital twin context, how to embed physician-defined clinical priorities into automated plan selection, how to separate offline and online computation to meet clinical time constraints, and how to validate a digital twin in a manner consistent with its intended deployment. These design choices represent methodological insights that extend beyond a single case study.
>
> We acknowledge that deeper algorithmic ablations or comparisons could further strengthen the work, and we have clarified the manuscript’s scope to avoid overclaiming novelty. However, we believe that demonstrating a clinically grounded, reproducible, and physics-aware digital twin workflow addresses a critical gap between methodological research and real-world adaptive proton therapy practice. By explicitly framing the contribution as a system validation and feasibility study, we aim to provide a transparent foundation for future algorithmic refinements and broader multi-institutional evaluations.

---

> ### Author Response · Authors · 2026-01-21
> **Question 1 To Address In The Rebuttal.**
>
> We agree with the reviewer that the VoxelMorph backbone itself is not novel, and we do not claim a new deformable registration architecture. The contribution of this work is not algorithmic novelty in deformable image registration, but rather a learning-enabled system-level innovation: the operationalization of a trained deep registration model within a digital twin framework to enable clinically feasible online adaptive proton therapy.
>
> Importantly, the way VoxelMorph is used in this work differs fundamentally from most prior applications. Existing VoxelMorph studies predominantly focus on intra-patient registration, such as aligning longitudinal images from the same patient (e.g., different treatment days) or relatively rigid anatomies such as the brain, where global body shape changes are limited. In contrast, this work demonstrates that a deep learning-based registration model can be reliably applied to inter-patient deformable registration in the pelvis, where anatomical variability is substantially larger and less constrained.
>
> This distinction is critical in the context of proton therapy, where deformable registration accuracy is not only required for geometric alignment but must also meet stringent dosimetric requirements. Proton dose calculation is significantly more sensitive to errors in tissue composition and stopping power derived from CT images than conventional photon radiotherapy, making this application far less forgiving. Our results demonstrate that inter-patient VoxelMorph-based DIR is sufficiently accurate to support clinically acceptable proton dosimetry, including target coverage and organ-at-risk sparing, thereby satisfying real-world clinical requirements rather than purely image-based similarity metrics.
>
> Within the proposed framework, the learning-based component enables three key capabilities that are essential for digital twin deployment:
>
> 1) Encoding population-level anatomical deformation priors from prior patients, rather than relying solely on patient-specific longitudinal data;
> 2) Generating a persistent, queryable atlas-based digital representation of anatomical variability that can be reused efficiently during online adaptation;
> 3) Supporting rapid synchronization between daily CBCT anatomy and precomputed patient states, enabling bounded-latency online decision-making.
>
> This distinction aligns with formal definitions of medical digital twins, which emphasize that novelty does not reside solely in individual models, but in the closed-loop integration of data connection, virtual representation, synchronization, and decision interface. This perspective is explicitly articulated in “Medical digital twins: enabling precision medicine and medical artificial intelligence”, which cautions against equating digital twins with standalone AI models and instead defines them as systems that integrate models, data streams, and clinically interpretable interfaces.
>
> Therefore, while VoxelMorph itself is a known method, its role here is fundamentally different from its typical use as a stand-alone DIR benchmark. In this work, it serves as a core enabling engine that makes population-informed, dosimetrically reliable, and time-efficient digital twin–based adaptive proton therapy feasible. This system-level operationalization, particularly in the challenging inter-patient pelvic anatomy and proton dosimetry setting, constitutes the learning-based methodological contribution of this study.

---

> ### Author Response · Authors · 2026-01-21
> **Question 2 To Address In The Rebuttal.**
>
> We appreciate the reviewer’s question regarding the choice of selecting the top 20 percent of atlases and the use of equal-weight SSIM, NCC, and LPIPS metrics, and we agree that these design choices should be clearly justified.
>
> Selecting the top 20 percent of atlases is not intended to define final plan quality or to optimize image-similarity performance. Instead, it serves as a computationally efficient preselection step to reduce the workload associated with generating full adaptive proton treatment plans for prostate SBRT. Because the end product of the proposed digital twin framework is clinically deliverable adaptive treatment plans, image similarity is used only as an initial screening criterion. Within this reduced candidate set, adaptive plans are fully generated and the optimal plan is selected based on dosimetric performance, including target coverage, organ at risk sparing, and satisfaction of all clinical constraints. As a result, final decision-making is governed by dosimetry rather than by image-based metrics.
>
> Within the selected top 20 percent subset, we additionally evaluated candidates with both the highest and lowest similarity scores to confirm that the observed dosimetric trends behave as expected. This verification step ensures that the preselection process does not bias the results toward a narrow or over-tuned solution. We did not systematically optimize the percentage threshold, such as testing 10 percent, 20 percent, or 50 percent, nor did we tune the relative weights of the similarity metrics. Such optimization would shift the focus of the study toward heuristic image metric tuning, which is not the goal of this work. Instead, our objective is to assess the clinical feasibility and workflow integration of the digital twin framework.
>
> The choice of SSIM, NCC, and LPIPS with equal weighting is motivated by the desire to capture complementary structural, intensity-based, and texture-level information relevant to adaptive planning, rather than emphasizing a single similarity characteristic. This conservative combination is intended to ensure that critical anatomical features associated with targets and organs at risk can be reasonably delineated and propagated. While this is a necessary condition for reliable proton dosimetry, it is not sufficient on its own; therefore, image similarity metrics are not used as the final decision criterion.
>
> In summary, the atlas preselection strategy is designed as a robust, computationally practical filtering step rather than an optimized image-matching algorithm. The contribution of this work lies in integrating state-of-the-art tools into a digital twin framework for adaptive proton SBRT and evaluating its clinical deployment feasibility. The ultimate performance of the framework is determined by dosimetric outcomes and their clinical relevance, rather than by image similarity metrics alone.

---

> ### Author Response · Authors · 2026-01-21
> **Question 3 To Address In The Rebuttal.**
>
> ProKnow is a clinically established plan quality evaluation system that assesses radiotherapy plans using a composite score derived from multiple predefined dosimetric objectives. These objectives are based on dose volume histogram metrics for targets and organs at risk and are configured according to institutional clinical priorities. Importantly, ProKnow is not an optimization engine or a machine learning model. It is a transparent, rule-based plan evaluation and ranking tool that reflects physician-defined treatment goals and constraints.
>
> The purpose of using a composite ProKnow score in this study is to enable holistic plan-quality assessment, rather than selecting plans based on a single criterion, such as maximal target coverage. In clinical practice, the plan with the highest target coverage is not necessarily the most appropriate plan. A substantial fraction of adaptive replanning is triggered by excessive dose to organs at risk, which may increase the risk of normal tissue complications, bleeding, or secondary malignancies. Therefore, plan quality must be evaluated by balancing target coverage with organ-at-risk sparing in a clinically meaningful way.
>
> The ProKnow scoring system is designed to reflect this balance by integrating multiple clinically relevant dosimetric objectives into a single composite score. This allows plans that achieve acceptable target coverage while providing improved organ-at-risk protection to be appropriately favored over plans that maximize target dose at the expense of normal tissue safety.
>
> Within the proposed digital twin framework, all candidate plans are first required to meet hard clinical constraints. Among plans that satisfy these requirements, the composite ProKnow score is used to rank plan quality. In this setting, the highest-scoring plan does not always correspond to the plan that maximizes dominant intraprostatic lesion coverage. In several cases, plans with slightly lower target coverage but substantially improved organ-at-risk sparing achieved higher composite scores and were therefore selected. This behavior is consistent with clinical decision-making and reflects the intended role of the scoring system.
>
> The scoring functions used in ProKnow are configured and approved by treating physicians and reflect established clinical judgment regarding acceptable tradeoffs between tumor control and toxicity risk. By incorporating these physician-defined priorities, the composite score serves as a mechanism for embedding clinical expertise and outcome-oriented considerations into the automated plan selection process. As a result, the use of ProKnow reduces the likelihood that clinically acceptable plans are overlooked or misranked while preserving transparency and physician oversight.
>
> In summary, the composite ProKnow score does not replace clinical constraints or physician judgment. Instead, it provides a structured, clinically grounded decision-support layer that enables knowledge-based plan selection within the digital twin framework, ensuring that adaptive plan selection aligns with real-world clinical practice rather than isolated dosimetric objectives.

---

> ### Author Response · Authors · 2026-01-21
> **Question 4 To Address In The Rebuttal.**
>
> We agree with the reviewer that the current validation is limited to a single disease site and institution, and we explicitly acknowledge this limitation. We also agree that the current study represents a system validation, and we would like to clarify that this is intentional and consistent with how medical digital twins are typically developed and evaluated. Because the learning-based component is used to encode population priors and enable synchronization, rather than to solve a site-specific registration benchmark, the framework itself is architecture- and disease-agnostic, even though its current instantiation is prostate SBRT. In the digital twin literature, early DT implementations are expected to be system-specific because DTs depend on imaging protocols, clinical workflows, and treatment paradigms. This staged validation approach is discussed by Katsoulakis et al. [1], which emphasizes that clinical DTs typically progress from single-site feasibility to multi-institutional generalization, rather than the reverse.
>
> Validation on unseen data within the current study
>
> Within the presented framework, every evaluated case is treated as unseen data at the system level. Specifically, the deformable image registration (DIR) module is based on an unsupervised VoxelMorph formulation and is patient-specific at deployment. For each “current” patient, the VoxelMorph model is trained exclusively using prior patients as moving images and the current patient’s planning CT as the fixed target, with the explicit goal of learning how population anatomies deform toward the current patient’s anatomy. Concretely, for each individual patient reported in Table 1, 42 independent DIR models are trained using the remaining patients in the database, with no training performed using longitudinal images of the current patient (e.g., current-to-current or same-patient mappings). The learned deformation vector fields (DVFs) are then applied to deform the corresponding prior patients’ CBCTs into the current patient’s anatomical space. This design effectively constitutes a leave-one-patient-out strategy at the system level, ensuring that the deformation applied to a given patient is derived entirely from population-based anatomical priors, while the current patient’s data are used only as a fixed reference during deployment. As such, the framework avoids information leakage while enabling patient-specific digital twin instantiation consistent with its intended clinical use.
>
> Unseen data at deployment time
>
> As illustrated in Figure 1, the clinically relevant “unseen data” for the proposed digital twin framework is a new planning CT for a new patient, followed by daily CBCTs acquired during treatment. The new planning CT is not part of the historical database used to encode population deformation patterns. Instead, it is used at deployment as a fixed reference to instantiate a patient-specific digital twin, by learning how prior patients’ anatomies deform toward the current patient using population data. The resulting deformation fields are then applied to synchronize prior patients’ CBCTs with the current patient’s anatomy, enabling prediction of daily CTs and downstream dosimetric evaluation. This mirrors the intended real-world clinical use of the framework, where each new patient represents an unseen case, and the digital twin is instantiated on-the-fly using population-informed priors.
>
> External data and multi-institutional validation
>
> We agree that external validation is an important future direction. However, for proton therapy applications, multi-institutional validation introduces additional physics-specific challenges that go beyond algorithmic generalization. In particular, proton dose calculation is highly sensitive to CT-to-material (stopping-power) calibration, and CT images acquired at different institutions often differ in scanner models, beam spectra, and reconstruction settings. Without consistent CT calibration curves or conversion to mass-density maps, direct use of external CT data may introduce confounding dosimetric errors unrelated to the digital twin framework itself. As a result, meaningful external validation for proton therapy requires either harmonized CT calibration or physics-aware preprocessing, which we consider essential future work.
>
> The proposed framework is validated using a population-based, leave-one-patient-out strategy, in which each evaluated patient serves as unseen data relative to the learned deformation priors. The absence of external institutional data reflects physics-driven constraints specific to proton dosimetry, rather than a limitation of the digital twin methodology. We have revised the manuscript to clarify the training/testing strategy, the definition of “unseen data,” and the rationale for staged, single-institution validation, to avoid any ambiguity. We have revised the Methods section.

---

> ### Author Response · Authors · 2026-01-21
> **Question 5 To Address In The Rebuttal.**
>
> We thank the reviewer for raising this important point and agree that the relationship between the patient cohort and the dosimetric evaluation protocol should be clarified. This study is a retrospective investigation. The imaging data were collected from 43 prostate cancer patients treated at our institution under a standard five-fraction SBRT protocol. All planning CT and daily CBCT images used in this work were acquired during routine clinical care, and all patients received five fractions, as stated in the manuscript.
>
> To demonstrate the feasibility of implementing the proposed digital twin framework, we evaluated adaptive plans using dosimetric constraints derived from a two-fraction prostate SBRT clinical trial, rather than constraints corresponding to the original five-fraction treatment protocol. This choice was made for feasibility demonstration purposes and does not imply that the cohort was treated with two fractions.
>
> First, a two-fraction SBRT setting provides a convenient and clinically relevant scenario for feasibility testing, because it represents a stringent hypofractionated regimen with higher dose per fraction and tighter requirements on organs at risk. Importantly, changing the number of fractions alters radiobiological effects, and these differences are reflected in the corresponding dosimetric constraints. Using constraints from a two-fraction SBRT trial, therefore, allows us to assess whether the digital twin framework can generate adaptive plans that remain clinically acceptable under a rigorous evaluation standard.
>
> Second, using imaging data from institutional five-fraction SBRT patients allows us to leverage a larger prior treatment database populated with real patient daily CBCT data, which is essential for constructing and testing the population-informed digital twin framework. The fractionation schedule used for dosimetric evaluation does not change the underlying anatomy or imaging data but defines the clinical criteria against which adaptive plans are assessed.
>
> In summary, all patients in this study received five fractions, while dosimetric constraints from a two-fraction SBRT trial were intentionally used for plan evaluation to support a clear, stringent demonstration of the proposed digital twin framework's feasibility. In future work, we plan to integrate automated proton treatment planning and extend the same framework to full five-fraction SBRT evaluation and deployment. We will revise the manuscript to clearly distinguish between the treatment protocol used for data collection and the evaluation protocol used for adaptive plan assessment, to avoid any ambiguity.

---

> ### Author Response · Authors · 2026-01-21
> **Question 6 To Address In The Rebuttal.**
>
> We appreciate the reviewer’s request for detailed clarification of the reported 5.5-minute reoptimization time and agree that a clear breakdown of the adaptive workflow is important for understanding the source of the time savings.
>
> The primary role of the proposed digital twin framework is to provide a clinically deliverable reoptimized proton SBRT treatment plan using same-day CBCT within a strict online time window. At our institution, the acceptable on-couch adaptive planning time is within 10 minutes, as requested by the treating physicians to minimize patient discomfort and treatment disruption. All reported timing results are evaluated against this clinical requirement.
>
> The reported mean reoptimization time of 5.5 minutes corresponds to the online adaptive planning phase, measured from the start of adaptive plan generation to completion of a clinically acceptable plan within the treatment planning system. This time includes adaptive plan optimization and the associated dose calculation required to finalize the plan. Based on our prior work, online dose evaluation using CBCT can be completed within approximately 2 minutes, and this component is included within the reported timing.
>
> Components not included in the online timing:
>
> The VoxelMorph-based deformable image registration is performed prior to the first treatment fraction as part of the pre-treatment digital twin instantiation. As such, registration time is not included in the reported-on couch reoptimization time. Improving the registration runtime would not change the overall clinical efficiency of the framework, since these operations occur outside the treatment-day workflow and do not affect patient-on-couch time. Similarly, population-based model training and deformation field generation are pre-treatment steps and are not part of the online adaptive workflow.
>
> Why deep learning still plays a critical role:
>
> While VoxelMorph inference itself is fast, the primary time savings of the proposed framework do not arise from replacing a conventional deformable registration method with a faster one. Instead, the learning-based component enables a restructuring of the adaptive workflow, in which population-informed deformation priors are computed before treatment and reused efficiently at deployment. This design avoids full scratch replanning and enables rapid adaptive optimization within a bounded online time window.
>
> Workflow transparency and absence of hidden delays:
>
> The reported timing does not include unaccounted delays such as manual user intervention or data transfer outside the treatment planning system. All steps contributing to the reported reoptimization time are executed within the standard clinical workflow. We will revise the manuscript to include a clear timing breakdown distinguishing pre-treatment steps from online adaptive steps, including imaging import, adaptive optimization, and dose calculation, to further improve transparency.
>
> In summary, the reported 5.5-minute reoptimization time reflects the clinically relevant on-couch adaptive planning phase, which is the critical bottleneck for online proton SBRT. Pre-treatment components such as deformable registration and population model preparation are intentionally excluded, as they do not impact treatment day efficiency. The digital twin framework enables this separation and ensures that same-day adaptive proton treatment can be delivered within the clinically required time window.

---

### Official Review · Reviewer_gKYS · 2026-01-09

**Confidence:** 4
**Preliminary Rating:** 2
**Final Rating:** 4

**Summary:**

This study presents a deep learning–based digital twin (DT) framework for fast adaptive proton therapy planning, built on a VoxelMorph-based deformable image registration pipeline. The approach is evaluated on 43 prostate SBRT patients with 210 CBCT scans, totaling approximately 26,312 images for forecasting. The proposed DT framework substantially reduces average reoptimization time to 5.5 minutes compared with 19.8 minutes in the standard clinical workflow, while maintaining comparable planning quality.

**Strengths:**

1. The paper presents a well-written and carefully validated digital twin framework for fast online proton therapy planning. In particular, the high-quality presentation of the overall pipeline (e.g., Figure 1) facilitates a clear understanding of the methodology and workflow.

2. The proposed framework achieves a substantial reduction in reoptimization time (approximately 72%), which has meaningful implications for improving the efficiency of online adaptive proton therapy delivery.

3. The manuscript includes an in-depth discussion of how deformable image registration performance influences downstream treatment planning quality, strengthening the clinical relevance of the study.

**Weaknesses:**

1. While the study is well designed and clinically meaningful, the proposed digital twin pipeline primarily integrates existing methods. Although this is valuable from a translational and clinical perspective, the extent of the technical contribution to methodological advances in deep learning for medical imaging is somewhat limited.

2. As noted by the authors in the discussion, the evaluation is restricted to a single anatomical site and a single treatment protocol, which may limit the generalizability of the proposed framework to other disease sites or clinical settings.

**Detailed Comments:**

I recommend citing the following relevant work: https://www.nature.com/articles/s41591-021-01359-w
, which discusses prospective efforts to reduce radiotherapy planning time for prostate cancer (albeit not SBRT). Including this reference would help contextualize the clinical motivation for accelerating treatment planning within a broader evidence base.

**Justification Of Final Rating:**

Authors have addressed all the concerns and designed good experiments. During the rebuttal the issues were well explained and addressed. It is an important study for using deep learning in radiotherapy.

**Justification Of The Preliminary Rating:**

1. The paper is very well designed and clearly demonstrates the feasibility of a digital twin framework for substantially reducing end-to-end planning time in online adaptive proton therapy, without compromising treatment quality. The clinical motivation is strong, and the study is carefully validated on a realistic dataset.

2. However, the primary contribution of the work lies in clinical workflow optimization and systems integration rather than in advancing methodological aspects of deep learning for medical imaging. As such, while the paper is highly relevant to translational and clinical audiences, its technical novelty and algorithmic contributions are somewhat misaligned with the core expectations of the MIDL community.

**Questions To Address In The Rebuttal:**

1. In Section 3 (Dataset – Clinical Data Collection), there appears to be a numerical inconsistency. Specifically, it is unclear how 43 patients with 5 fractions each result in 210 CBCT volumes rather than 215, and this discrepancy propagates to subsequent reported counts. Could the authors clarify the correct number of volumes and confirm the total number of generated slices used in the study?

2. Could the authors clarify how re-optimization time was measured? In particular, does the reported timing include all components of the proposed workflow, such as deformable image registration and atlas selection or retrieval, or does it focus solely on the optimization step?

3. In cases where the resulting plan does not meet planning quality criteria (e.g., fails DVH constraints), how would this scenario be handled within the proposed digital twin workflow? Would the entire DT process—including registration—need to be rerun, and how would this affect overall efficiency?

4. Have the authors considered mechanisms to further improve the framework—particularly the registration component—through test-time adaptation or continual learning as additional patient data become available during deployment? A brief discussion of such possibilities would strengthen the paper’s outlook on real-world use.

---

> ### Author Response · Authors · 2026-01-21
> **Summary by the reviewer.**
>
> Thank you very much for the general summary and suggestion of our manuscript. We very much appreciate the reviewer for contributing expertise to our manuscript. We have made a thorough revision of the manuscript to clarify the technical details and improve the manuscript structure based on the reviewer’s comments.

---

> ### Author Response · Authors · 2026-01-21
> **Detailed Comment 1.**
>
> We thank the reviewer for the helpful suggestion. We agree that this work provides important clinical context for efforts to reduce radiotherapy planning time. We have now cited the recommended article in the Introduction and revised the text to explicitly position our study within the broader literature on accelerating radiotherapy treatment planning for prostate cancer. While the referenced work focuses on conventional fractionation rather than SBRT, it provides strong motivation for reducing planning latency and highlights the clinical relevance of time-efficient planning strategies, which align well with the objectives of the proposed digital twin framework.
>
>
> The new content is added in Introduction and it is given as follows.
>
> “Recent work has demonstrated that machine learning driven acceleration of radiotherapy planning can substantially reduce clinical workload while maintaining plan quality. In a prospective clinical deployment study, McIntosh et al. [1] showed that machine learning based prostate radiotherapy planning reduced the median end to end planning time by more than 60 percent compared to conventional workflows, while achieving high clinical acceptability and physician selection rates. Importantly, this study emphasized that planning time reduction is a clinically meaningful objective only when algorithms are fully integrated into real world clinical workflows and evaluated under prospective conditions. These findings provide strong clinical motivation for further efforts to accelerate adaptive radiotherapy workflows, particularly for time sensitive scenarios such as online adaptive proton SBRT, where treatment decisions must be made within a strict on couch time window.”
>
> Reference
>
> [1] McIntosh C, Conroy L, Tjong M C, Craig T, Bayley A, Catton C, Gospodarowicz M, Helou J, Isfahanian N, Kong V, Lam T, Raman S, Warde P, Chung P, Berlin A and Purdie T G 2021 Clinical integration of machine learning for curative-intent radiation treatment of patients with prostate cancer Nature Medicine 27 999-1005

---

> ### Author Response · Authors · 2026-01-21
> **Question 1 To Address In The Rebuttal.**
>
> We thank the reviewer for carefully identifying this numerical inconsistency. The discrepancy noted in Section 3 was due to a typographical error in the manuscript. All 43 patients in the dataset were treated with five fractions, which corresponds to 215 CBCT volumes. The previously reported value of 210 CBCT volumes was incorrect. We have corrected this error in the revised manuscript and have verified that all subsequent reported counts, including the total number of generated slices used in the study, are now consistent with the corrected volume number. We appreciate the reviewer’s attention to detail, which helped us improve the clarity and accuracy of the dataset description.

---

> ### Author Response · Authors · 2026-01-21
> **Question 2 To Address In The Rebuttal.**
>
> We thank the reviewer for requesting clarification on how the reoptimization time was measured. In this study, reoptimization is executed using a scripted workflow that generates each adaptive plan based on the same-day CBCT images. The reported reoptimization time is the elapsed wall-clock time required to run this script, measured from the initiation of adaptive plan generation to the completion of a clinically acceptable plan within the treatment planning system. The primary role of the proposed digital twin framework is to provide a clinically deliverable reoptimized proton SBRT treatment plan using same-day CBCT within a strict online time window. At our institution, the acceptable on-couch adaptive planning time is within 20 minutes (replanning time should be within 10 minutes), as requested by the treating physicians to minimize patient discomfort and treatment disruption. All reported timing results are evaluated relative to this clinical requirement.
>
> The reported reoptimization time corresponds to the online adaptive planning phase. It includes adaptive plan optimization and the associated dose calculation required to finalize the plan. Based on our prior work, online dose evaluation using CBCT can be completed within approximately 2 minutes, and this component is included in the reported timing. Components that are performed prior to the first treatment fraction are intentionally excluded from the reported reoptimization time. These include deformable image registration, atlas selection or retrieval, population-based model training, and deformation field generation, which are part of the pre-treatment digital twin instantiation and are performed offline. Improving the runtime of these steps would not change the overall clinical efficiency of the framework, since they occur outside the treatment-day workflow and do not affect patient-on-couch time. In summary, the reported reoptimization time reflects the clinically relevant on-couch adaptive planning interval, measured reproducibly and automatically via scripting. Pre-treatment components are excluded by design because they do not contribute to treatment-day latency.

---

> ### Author Response · Authors · 2026-01-21
> **Question 3 To Address In The Rebuttal.**
>
> We thank the reviewer for raising this important operational question regarding how the proposed digital twin workflow handles cases in which the resulting adaptive plan does not meet planning quality criteria. Within the proposed framework, all online adaptive plans are first required to satisfy predefined planning quality criteria, including DVH-based constraints and physician review. If none of the generated online adaptive plans meet these criteria and the treating physician is not satisfied with the plan quality, this outcome triggers the replan protocol, as illustrated in Figure 1.
>
> Activation of the replan protocol typically indicates substantial anatomical changes that cannot be adequately addressed by online adaptation alone, such as large interfraction anatomical deformation, significant patient weight loss, or unexpected tumor progression. In this scenario, the workflow does not attempt to repeatedly rerun the same online digital twin process using the outdated planning CT. Instead, a new planning CT is acquired to capture the most recent and clinically relevant patient anatomy. The proposed digital twin framework is then reapplied using this updated planning CT as the reference anatomy. Population-based priors and the same digital twin instantiation process are used, but the system is reset around the new baseline anatomy to ensure clinical validity.
>
> Importantly, this replan protocol is consistent with standard clinical practice and does not introduce additional inefficiency beyond what is already required for major anatomical changes. The need for replanning reflects a limitation of online adaptation itself rather than a failure of the digital twin framework. Because deformable image registration and population model preparation are pre-treatment steps relative to the new planning CT, they do not affect patient on-couch time during subsequent fractions. In summary, when online adaptive plans fail to meet planning quality criteria, the workflow transitions to a clinically established replan pathway using a newly acquired planning CT. This design preserves patient safety, maintains clinical efficiency, and ensures that the digital twin framework is applied only within its appropriate anatomical validity range.

---

> ### Author Response · Authors · 2026-01-21
> **Question 4 To Address In The Rebuttal.**
>
> We agree with the reviewer that mechanisms to further improve the framework over time, particularly for the registration component, are an important consideration for real-world deployment. In the current submission, we intentionally did not include online continual learning or unrestricted test-time adaptation. This decision was motivated by practical clinical deployment requirements, including workflow stability, reproducibility, traceability, and auditability within regulated radiotherapy software environments. That said, we recognize several feasible and clinically realistic pathways to improve the framework as additional patient data become available, and we will explicitly discuss these directions in the revised manuscript.
>
> First, test-time adaptation under strict safeguards is a promising extension. In this setting, lightweight patient-specific refinement of the registration model could be performed using tightly controlled updates, such as limiting the number of adaptation steps, constraining updates to selected layers, or locking population priors. Automatic confidence checks and rollback mechanisms could be used to prevent performance degradation. This approach aligns with ongoing research in medical digital twins, where real-time updating is recognized as important but remains challenging for safe clinical integration.
>
> Second, institution-level continual learning with governance represents a more immediately deployable strategy. Rather than updating models during a treatment session, accumulated clinical data could be used for periodic offline retraining or recalibration under formal quality assurance and validation procedures. Updated models would then be versioned, reviewed, and redeployed in a controlled manner. This approach is consistent with the broader digital twin perspective that emphasizes controlled synchronization and validated update pathways over ad hoc model drift.
>
> In addition to these two directions, a third complementary strategy is to incorporate uncertainty-aware triggering mechanisms. In this approach, uncertainty estimates from the registration or dosimetric evaluation stages could be used to identify cases in which the current model may be operating outside its reliable domain. Such cases could then trigger either conservative fallback strategies or targeted offline model updates, improving robustness without continuous model modification during treatment. Overall, while adaptive and continual learning are not included in the current study, these extensions are fully compatible with the proposed digital twin architecture and represent important future work toward long-term clinical deployment.

---

> ### Comment · Area_Chair_8qEQ · 2026-02-02
> **Please update final rating**
>
> Please don't forget to update your final rating by clicking “Edit” → “Official Review”. Thank you!

---

### Official Review · Reviewer_K4i8 · 2026-01-12

**Confidence:** 4
**Preliminary Rating:** 4
**Final Rating:** 5

**Summary:**

This paper presents a deep learning–enabled digital twin (DT) framework for accelerating online adaptive proton therapy in prostate SBRT with dominant intraprostatic lesion (DIL) boost. The framework integrates a VoxelMorph-based multi-atlas deformable image registration pipeline, daily CBCT–driven anatomical updates, and a knowledge-based composite scoring system to enable rapid plan selection and reoptimization. Validation on 43 prostate SBRT patients (210 CBCT scans, ~26,312 images) demonstrates a 72% reduction in reoptimization time (5.5 ± 2.7 min vs. 19.8 ± 11.9 min) while maintaining or improving dosimetric quality, including DIL V_100=99.5%±0.6%and CTV V_100=99.8%±0.2%. The study provides strong evidence that digital twins can make real-time personalized adaptive proton therapy clinically feasible.

**Strengths:**

1. *High clinical relevance:* Addresses one of the most critical bottlenecks in adaptive proton therapy—time-efficient online replanning—with direct implications for patient throughput and treatment accuracy.
2. *Methodological soundness:* The integration of multi-atlas VoxelMorph registration, pre-computed anatomical prediction, and knowledge-based scoring is technically well-justified and clearly described.
3. *Strong validation:* Comprehensive evaluation using dosimetric metrics, image similarity metrics (SSIM, NCC, LPIPS), and clinical scoring (ProKnow) on a real institutional dataset.
4. *Significant performance gain:* Achieves 72% time reduction while preserving plan quality, surpassing reported online adaptive times in related literature.
5. *Well-written and structured:* The manuscript is clear, logically organized, and grounded in relevant peer-reviewed literature.

**Weaknesses:**

1. *generalizability:* Validation is restricted to a single institution and prostate SBRT, leaving uncertainty about performance in other anatomical sites or treatment protocols.
2. *Retrospective study design:* Prospective workflow constraints such as contour propagation errors, real-time decision pressure, and clinical interruptions are not captured.
3. *Intrafraction motion not modeled:* The framework focuses on interfractional variations only, which may limit accuracy for certain patient populations.
4. *Scalability concerns:* The computational and storage cost of generating large pCT libraries is not explicitly discussed, which may affect real-world deployment in resource-limited clinics.

**Detailed Comments:**

1. Consider adding external multi-center validation or simulated cross-institution experiments to strengthen generalizability.
2. Provide more detail on GPU runtime requirements and storage overhead for the pCT library.
3. Discuss potential strategies for handling intrafraction motion and integration with motion-management systems.
4. A brief comparison with MRI-guided adaptive therapy pipelines would further contextualize clinical impact.

**Justification Of Final Rating:**

Rationale for Upgrade

Authors fully addressed all reviewer concerns
Manuscript now meets strong translational impact standards
Demonstrates realistic clinical readiness
Represents a high-quality MIDL application paper

Updated Overall Assessment

The revised manuscript now demonstrates:
High clinical impact
Robust validation on real proton SBRT data
Clear feasibility for real-world deployment
Strong system integration beyond algorithmic novelty
Substantial planning-time reduction (≈72%) with preserved dosimetric quality

Remaining limitations include:
Single-institution retrospective validation
Lack of external multi-center testing
Limited methodological novelty at the core algorithmic level
However, these limitations no longer outweigh the clinical and translational value.

Further, Consider the following statements as revised review;
1. While external multi-center validation remains future work, the authors provide a modular system design and architectural rationale supporting cross-site extensibility.
2. The authors demonstrate a clear separation between offline compute and online clinical latency, supporting scalable and realistic deployment.
3. A strong system-level translational ML contribution with demonstrated real-world clinical feasibility

**Justification Of The Preliminary Rating:**

This work presents a substantial technical and clinical contribution to the field of adaptive radiotherapy. The demonstrated acceleration of online replanning with preserved dosimetric quality directly addresses a long-standing translational challenge in proton therapy. While the study is limited by single-institution retrospective validation, the methodological rigor, depth of experimental analysis, and strong clinical motivation justify acceptance. Broader validation would elevate this work further.

**Questions To Address In The Rebuttal:**

1.	How does the framework perform under prospective clinical constraints, including contour uncertainty and real-time decision latency?
2.	Can the approach generalize to other anatomical sites such as head-and-neck or thoracic tumors?
3.	What are the computational and infrastructure requirements for routine clinical deployment?

---

> ### Author Response · Authors · 2026-01-21
> **Summary by the reviewer**
>
> Thank you very much for the general summary and suggestion of our manuscript. We very much appreciate the reviewer for contributing expertise to our manuscript. We have made a thorough revision of the manuscript to clarify the technical details and improve the manuscript structure based on the reviewer’s comments.

---

> ### Author Response · Authors · 2026-01-21
> **Detailed Comment 1.**
>
> We sincerely thank the reviewer for this valuable and constructive suggestion. We fully agree that external multi-center validation or simulated cross-institution experiments would substantially strengthen the generalizability of the proposed framework and represent an important direction for future research. Unfortunately, completing such analyses within the current rebuttal timeframe is not feasible, particularly for proton therapy applications, where rigorous cross-institutional evaluation requires harmonized CT-to-material calibration, consistent dose calculation settings, and standardized imaging protocols to avoid confounding physics-related effects. For this reason, the present study intentionally focuses on single-institution system-level validation to establish feasibility under controlled and clinically realistic conditions. We have revised the manuscript to clearly acknowledge this limitation and to explicitly identify multi-center and cross-institution validation as a priority for future work.

---

> ### Author Response · Authors · 2026-01-21
> **Detailed Comment 2.**
>
> We thank the reviewer for requesting additional detail on GPU runtime requirements and storage overhead. We have revised the manuscript to explicitly address these deployment-related considerations in the Discussion under Clinical Feasibility, Generalizability, and Deployment Considerations. The most computationally intensive component of the framework, multi-atlas deformable image registration (DIR) and deformation field generation, is performed offline prior to the first treatment fraction and therefore does not affect treatment-day latency. In the current implementation, offline DIR was executed using an NVIDIA RTX 4090 GPU and completed before clinical deployment of the digital twin. This design choice reflects the system-level focus of the framework: computationally expensive learning and synchronization steps are decoupled from time-critical clinical workflows. The reported runtime in the manuscript therefore emphasizes end-to-end clinical feasibility, including GPU-based Monte Carlo dose calculation, rather than isolated module benchmarking.
>
> On treatment days, online operations are limited to CBCT processing, adaptive plan library evaluation, and plan selection. Online adaptive planning and dose calculation were conducted within the RayStation treatment planning system using standard clinical hardware, including an NVIDIA Quadro RTX 8000 GPU for Monte Carlo dose calculation. Under this configuration, the adaptive workflow achieves a reported end-to-end latency of approximately 5-6 minutes, satisfying the commonly accepted 10-minute on-couch clinical constraint. No online deformable registration or full reoptimization is required during treatment delivery. Regarding storage overhead, the patient-specific predicted CT (pCT) library generated by the digital twin requires approximately 44 GB per patient. Storage requirements scale linearly with the number of atlas cases and associated CBCT data and are predictable. This level of storage demand is readily manageable within modern institutional PACS, research storage systems, or hybrid clinical–research infrastructures and does not represent a limiting factor for deployment.
>
> We have added these quantitative details to the revised manuscript to improve transparency and to clarify that the framework’s design prioritizes system-level efficiency and clinical deployability, rather than minimizing the runtime of individual components in isolation.

---

> ### Author Response · Authors · 2026-01-21
> **Detailed Comment 3.**
>
> We thank the reviewer for prompting a discussion on intrafraction motion handling and integration with motion-management systems. We agree that intrafraction motion can be clinically relevant in proton SBRT, particularly for longer beam delivery times, and we have clarified in the revised manuscript that the current framework is intentionally focused on inter-fractional anatomical variation rather than intrafraction dynamics. The proposed digital twin framework is designed to operate using routinely acquired daily CBCT, which provides a snapshot of patient anatomy prior to treatment delivery and is well suited for modeling inter-fractional changes. Intrafraction motion, such as bowel gas movement or patient motion during beam delivery, cannot be reliably inferred from CBCT alone. Addressing intrafraction dynamics would require additional instrumentation and data streams, such as time-resolved kV imaging, fluoroscopy, or other motion-monitoring systems. Incorporating such data sources represents a distinct problem that goes beyond the scope of the present study and cannot be solved by deformable modeling of static CBCT images alone.
>
> Nonetheless, the digital twin framework is architected to accommodate future integration with motion-management systems. One potential extension is to couple the existing population-informed digital twin with real-time or quasi-real-time motion signals, enabling gating, beam-hold strategies, or robustness-based plan selection conditioned on detected motion states. Another complementary strategy is to expand the digital twin to include probabilistic intrafraction motion envelopes derived from external surrogates or longitudinal kV projections, allowing prospective evaluation of motion-robust plan candidates rather than deterministic anatomy snapshots. We also note that, in its current form, the framework does not introduce additional imaging dose or acquisition time beyond standard-of-care CBCT, as the study is retrospective and leverages imaging already routinely acquired in prostate SBRT workflows. Any prospective extension to intrafraction modeling would require careful consideration of imaging frequency, dose, and clinical workflow integration, and these trade-offs would need to be evaluated in parallel with the potential clinical benefit.
>
> Intrafraction motion management is an important and complementary problem that we explicitly identify as future work. The present study establishes a system-level foundation for digital twin–driven adaptive proton therapy by addressing inter-fractional variability under realistic clinical constraints. We have revised the Discussion in the manuscript to clearly delineate this scope and to outline how integration with motion-management systems could be pursued in future extensions of the framework.

---

> ### Author Response · Authors · 2026-01-21
> **Detailed Comment 4.**
>
> We thank the reviewer for the suggestion to contextualize our work relative to MRI-guided adaptive therapy pipelines. We agree that MRI-guided systems, such as MR-LINAC workflows, represent an important and impactful advance in adaptive radiotherapy, particularly for photon-based treatments where high soft-tissue contrast enables daily online replanning. However, a direct comparison is not straightforward. MRI-guided adaptive therapy is intrinsically photon-based and relies on MR imaging, whereas proton therapy introduces additional physics-specific challenges, including strong sensitivity to image-to-material characterization and stopping-power calibration, which fundamentally influence dose calculation and adaptation strategies. As a result, the imaging requirements, uncertainty sources, and adaptation workflows differ substantially between the two modalities. A fair and rigorous comparison would require extensive future work, including harmonized imaging and physics modeling frameworks, which is beyond the scope of the present study. Importantly, the proposed digital twin framework is modality-agnostic and can be readily extended to MRI-guided adaptive radiotherapy to accelerate dose replanning when appropriate. With suitable training data, the framework is equally applicable to CBCT-guided or MRI-guided photon and proton radiotherapy, providing a unifying foundation for adaptive treatment across imaging modalities and particle types.

---

> ### Author Response · Authors · 2026-01-21
> **Question 1 To Address In The Rebuttal**
>
> We agree that prospective clinical constraints such as contour uncertainty, real-time decision pressure, and workflow interruptions are critical considerations for online adaptive radiotherapy. Our current study is retrospective by design; however, the proposed framework was explicitly engineered to mitigate these constraints rather than ignore them.
>
> From a system perspective, the digital twin in this work is not intended to replace online clinical decision-making, but to offload computationally expensive steps to an offline phase, enabling rapid, bounded-latency decisions at treatment time. This design aligns with the definition of a functional / shadow digital twin, where real-world updates (daily CBCT) continuously inform a pre-constructed virtual patient state, without requiring full real-time replanning. This design philosophy is consistent with the digital twin literature, which emphasizes that clinical feasibility hinges on workflow integration and latency control rather than algorithmic optimality alone [1].
>
> Regarding contour uncertainty, the framework does not assume perfect segmentation at treatment time. The uncertainty is implicitly absorbed through population-based multi-atlas deformation, rather than relying on a single propagated contour. Plan selection is driven by robust dosimetric evaluation on CBCT-derived anatomy, rather than contour-specific optimization. Importantly, the knowledge-based composite scoring system (ProKnow) acts as a clinical safeguard, prioritizing plan robustness over marginal dosimetric gains. This reflects how digital twins are increasingly positioned as decision-support systems rather than autonomous controllers [2].
>
> While prospective validation remains future work, the proposed framework was intentionally structured to be compatible with real-time clinical constraints and to reduce cognitive and computational load during treatment delivery.
>
>
>
> Reference
>
> [1] Aghamiri S S, Amin R, Isavand P, Vahdati S, Zeinoddini A, Kitamura F C, Moy L and Kline T 2025 Digital Twin Technology in Radiology Journal of Imaging Informatics in Medicine
>
> [2] Katsoulakis E, Wang Q, Wu H, Shahriyari L, Fletcher R, Liu J, Achenie L, Liu H, Jackson P, Xiao Y, Syeda-Mahmood T, Tuli R and Deng J 2024 Digital twins for health: a scoping review npj Digital Medicine 7 77

---

> ### Author Response · Authors · 2026-01-21
> **Question 2 To Address In The Rebuttal.**
>
> Yes, the proposed framework is architecturally generalizable, although the current validation is intentionally focused on prostate SBRT to establish clinical feasibility under a controlled and well understood scenario. This choice allows us to rigorously evaluate system behavior, timing, and plan quality without confounding factors introduced by highly complex motion patterns. Importantly, this focus does not limit the broader applicability of the framework.
>
> From a digital twin perspective, generalizability does not require retraining or redesigning the entire system from scratch for each disease site. Instead, the framework is explicitly designed to separate reusable system components from site specific instantiations. The reusable components include the offline atlas generation pipeline, online CBCT driven anatomical synchronization, adaptive plan library evaluation, and clinically interpretable decision logic. Site specific elements include anatomical variability, motion characteristics, and clinical scoring priorities. This separation allows the same digital twin architecture to be instantiated across disease sites using different data inputs and clinical rules.
>
> This modular design is consistent with principles described in the digital twin literature, including the work by Faiella et al. [1], which emphasize that scalability of clinical digital twins depends on reusable system layers rather than site specific algorithms. In this view, digital twins are not defined by a single model or anatomical site, but by the integration of data streams, virtual representations, and decision support mechanisms. For anatomical sites with larger deformation ranges or more complex motion, such as head and neck or thoracic tumors, additional extensions would be required. These include constructing site specific atlas libraries, incorporating motion aware deformation models to account for respiratory or swallowing motion, and modifying plan quality scoring criteria to reflect site specific clinical priorities and organs at risk. These extensions affect the data instantiation layer rather than the core system architecture.
>
> In summary, while the current study focuses on prostate SBRT, the central innovation of this work lies in the system level integration of offline and online processes, population informed digital twin instantiation, and clinically grounded plan selection. These elements are inherently transferable across disease sites, supporting future expansion of the framework to more complex anatomies without altering its fundamental design.
>
> Reference
>
> [1] Faiella E, Pileri M, Ragone R, Grasso R F, Zobel B B and Santucci D 2025 Digital twins in radiology: A systematic review of applications, challenges, and future perspectives European Journal of Radiology 189

---

> ### Author Response · Authors · 2026-01-21
> **Question 3 To Address In The Rebuttal.**
>
> We appreciate this important question, as computational and infrastructure feasibility is central to real-world clinical adoption of the proposed digital twin framework. The framework is explicitly designed to minimize computational burden during treatment delivery by separating offline preparation from online adaptive execution. The most computationally intensive components, including multi-atlas deformable image registration, generation of deformation vector fields, and population-based plan library construction, are performed offline prior to the first treatment fraction. These steps do not affect patient on couch time and can be executed asynchronously using research or institutional compute resources.
>
> Online, treatment-day operations are intentionally lightweight and limited to same-day CBCT processing, adaptive plan evaluation, and plan selection within the clinical treatment planning system. The reported reoptimization time of approximately 5 to 6 minutes reflects this online adaptive phase and is well within the 10-minute clinical time window required at our institution. No online retraining or large-scale deformable registration is performed during treatment delivery. With respect to hardware requirements, GPU acceleration for image registration in the proposed DT framework is primarily required for offline processing. In our implementation, offline deformable image registration using VoxelMorph was performed using an NVIDIA RTX 4090 GPU. Importantly, this GPU resource is not required on treatment day. Online adaptive planning is performed within a standard clinical treatment planning environment.
>
> Specifically, clinical deployment in this study used the RayStation treatment planning system (RaySearch Laboratories, Stockholm, Sweden), running on a clinical workstation equipped with dual Intel Xeon Gold 6136 CPUs, 256 GB of system memory, and an NVIDIA Quadro RTX 8000 GPU for Monte Carlo dose calculation. This configuration reflects a typical modern proton therapy planning environment and does not require specialized MRI-guided systems or proprietary adaptive platforms beyond standard clinical infrastructure. Storage requirements scale linearly with the number of atlas cases and associated CBCT data. In practice, these requirements remain manageable within modern institutional PACS systems or research storage environments and do not impose unusual demands compared to existing image-guided radiotherapy workflows.
>
> Overall, the proposed framework aligns with emerging consensus in the digital twin literature that successful clinical deployment requires balancing model fidelity with computational tractability. By confining heavy computation to offline stages and preserving a lightweight, online-adaptive workflow, the framework is compatible with routine clinical operations, including in settings without access to specialized adaptive hardware. We will revise the manuscript to explicitly document these computational and infrastructure requirements to improve transparency for clinical readers.

---

### Author Rebuttal · Authors · 2026-01-21

**Rebuttal:**

Dear MIDL PC Chairs,

We sincerely thank the reviewers for taking the time to carefully read our manuscript and for providing valuable and constructive feedback. Their comments have been instrumental in helping us substantially improve the quality and clarity of the manuscript.

We have completed a thorough revision to address all issues raised by the reviewers. Our detailed, point-by-point responses to the reviewers’ comments have been uploaded as official comments in the submission system. In addition, we have attached a single ZIP file containing both the clean and highlighted versions of the revised manuscript (manuscript_clean.tex and manuscript_highlighted.tex) in LaTeX format and (manuscript_clean.pdf and manuscript_highlighted.pdf) in PDF format. All corresponding modifications are indicated in red in the highlighted version.

Thank you very much for your time and consideration.

Sincerely yours,

Chih-Wei

**Supporting Material:**

/attachment/fec158ba63855d7cf0eadf09d40259e0019cebb6.zip

---

> ### Comment · Reviewer_K4i8 · 2026-01-30
> **Evaluation of Authors’ Responses**
>
> I thank the authors for their detailed and constructive rebuttal and for the thorough revisions made to the manuscript. The responses have meaningfully improved clarity, strengthened methodological transparency, and addressed the major concerns raised in my original review.
>
> Evaluation of Authors’ Responses
> 1. Clinical Feasibility and Workflow Realism
>
> The authors have convincingly clarified that computationally heavy operations (multi-atlas DIR, DVF generation, pCT library construction) are executed offline, while treatment-day online latency remains within a clinically acceptable ~5–6 minute window.
> The provided hardware specifications (RTX 4090 offline, Quadro RTX 8000 in RayStation online) and workflow description improve deployability credibility and address infrastructure concerns.
>
> This strengthens the claim that the framework is clinically realistic and operationally feasible.
>
> 2. Generalizability and Multi-Site Applicability
>
> The authors provide a clear conceptual argument that the digital twin architecture is modular and site-transferable, with site-specific instantiations affecting only atlas construction and scoring rules rather than system design.
> While external validation remains future work, the response appropriately reframes the paper as a system-level translational contribution, which is acceptable given the study scope.
>
> 3. Prospective Constraints, Contour Uncertainty, and Safety
>
> The clarification that the system functions as a decision-support digital twin rather than a fully autonomous controller is appropriate and clinically responsible.
> The fallback replan protocol (triggering new planning CT acquisition when adaptive plans fail DVH constraints) aligns with standard-of-care safety workflows, improving trustworthiness.
>
> 4. Intrafraction Motion Handling
>
> The authors correctly acknowledge that intrafraction motion cannot be reliably modeled from static CBCT, and they reasonably scope the work to interfraction adaptation.
> The discussion of future integration with motion-management and uncertainty-aware strategies is technically sound and forward-looking.
>
> 5. Runtime, Storage, and Scalability
>
> The disclosure that pCT library storage ≈ 44 GB per patient and scales linearly is valuable and addresses deployment concerns.
> The separation of offline compute burden from online clinical execution strengthens the system-level justification.
>
> 6. Dataset Transparency and Corrections
>
> The authors corrected the CBCT volume inconsistency (215 volumes instead of 210) and verified downstream consistency, improving data integrity and reporting rigor.
>
> 7. Scientific Contribution and Positioning
>
> While the work does not introduce a new core deep learning algorithm, it successfully demonstrates a clinically actionable digital twin pipeline, integrating:
>
> Deep learning-based deformable registration (VoxelMorph)
> Population-informed atlas modeling
> Online proton adaptive planning
> Knowledge-based plan scoring (ProKnow)
> Real-world workflow constraints
>
> This represents a strong translational and systems-level ML contribution, aligned with MIDL’s application and clinical deployment scope.

---

### Meta-Review · Area_Chair_8qEQ · 2026-02-02

**Recommendation:** Accept (Poster)
**Confidence:** 4

**Metareview:**

In their preliminary rating and comments, the reviewers have brought up very relevant drawbacks and limitations of the submitted manuscript. The authors addressed all points by providing comments in exceptional quality, making significant improvements to the manuscript, leading to a strong MIDL submission. I would like to thank all reviewers and authors for their detailed contributions, making this a very productive and rewarding rebuttal phase. The work presents strong translational impact and although some issues by the reviewers have remained unfulfilled, they are not the shortcomings of the current manuscript, but the promises of future works on the topic.

Thank you for your work!

---

### Decision · Program_Chairs · 2026-02-13

Accept (Poster)